# SSRL: Self-Search Reinforcement Learning

## Abstract

We investigate the potential of large language models (LLMs) to serve as efficient simulators of world knowledge for agentic search tasks in reinforcement learning (RL), thereby reducing dependence on costly interactions with external search engines. To this end, we first quantify the intrinsic search capability of LLMs via structured prompting and repeated sampling, which we term Self-Search. Our results reveal that LLMs exhibit strong scaling behavior with respect to the inference budget, achieving high `pass@k` on QA benchmarks, including the challenging BrowseComp task. Building on these observations, we introduce Self-Search RL (SSRL), which enhances LLMs' Self-Search capability through format-based and rule-based rewards and enables models to iteratively refine their knowledge utilization internally, without requiring access to external tools. Empirical evaluations demonstrate that SSRL-trained models provide a cost-effective and stable environment for search-driven RL training, reducing reliance on external search engines and facilitating robust sim-to-real transfer. We draw the following conclusions: 1) LLMs possess world knowledge that can be effectively elicited to achieve high performance; 2) SSRL demonstrates the potential of leveraging internal knowledge to reduce hallucination; 3) SSRL-trained models integrate seamlessly with external search engines without additional effort. Our findings highlight the potential of LLMs to support more scalable RL agent training.

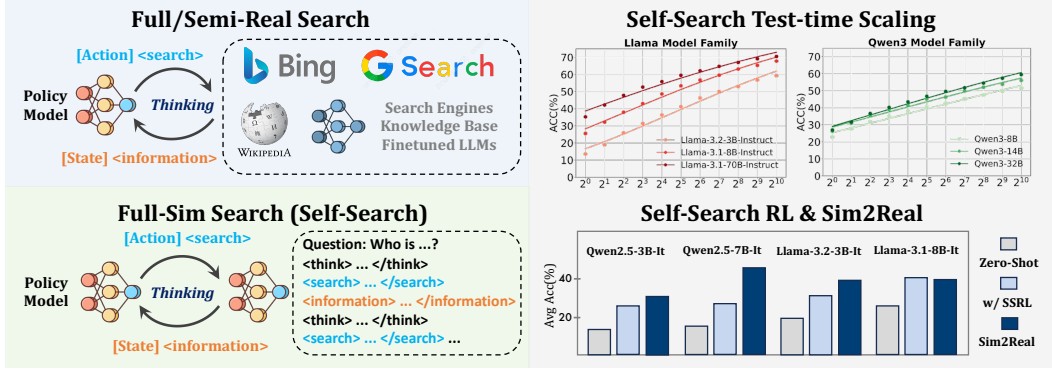

Figure 1: Left: Prior methods like Search-R1 (Jin et al., 2025b) and ZeroSearch (Sun et al., 2025) rely on external sources (e.g., search engines, knowledge bases, or fine-tuned LLMs), representing full or semi-real search. We propose full-sim search, where a policy model generates entire rollouts internally, including reasoning, searching and information (**Self-Search**). Right: Self-Search with test-time scaling shows strong `pass@k` performance as compute increases. Self-Search Reinforcement Learning (**SSRL**) further boosts results across models and tasks, especially with sim-to-real generalization.

## 1 Introduction

Recently, Reinforcement Learning (RL) with verifiable rewards has substantially improved the reasoning abilities of Large Language Models (LLMs) in complex mathematical problem-solving (OpenAI, 2024; DeepSeek-AI, 2025; Petrov et al., 2025) and code generation (El-Kishky et al., 2025; Cui

et al., 2025), leading to the emergence of Large Reasoning Models (LRMs) (Xu et al., 2025). Beyond mathematics and coding, numerous studies have explored the application of RL to LLMs in agentic contexts such as tool learning (Qian et al., 2025; Feng et al., 2025). These approaches enable LLMs to learn to invoke external tools such as web search engines, perform actions, and observe states within real-world environments. Although recent models like Search-R1 (Jin et al., 2025b) and Kimi V2 (Team et al., 2025) have achieved strong performance on various benchmarks, interacting with real web search engines remains costly (Sun et al., 2025), especially given the large number of rollouts and multi-turn tool calls required during RL training. In fact, due to pre-training on massive web-scale corpora (Brown et al., 2020; Liu et al., 2024; Yang et al., 2025b), LLMs can often answer questions involving world knowledge. Some studies also suggest that LLMs can serve as world models by providing state information in response to given actions (Li et al., 2023; Hao et al., 2023; Gu et al., 2024; Tang et al., 2024). For example, recent work on ZeroSearch (Sun et al., 2025) demonstrates that a fine-tuned LLM can effectively replace web search, providing stable and reliable knowledge. This finding indicates that the cost of search RL can be significantly reduced by adopting a semi-real setting. Inspired by recent advances in unsupervised RL like TTRL (Zuo et al., 2025), we explore self-search RL within fully simulated RL settings (noted as *full-sim*), where no real search is used during training. Specifically, we focus on two key research questions: 1) *What is the performance limit of LLMs on search-based QA tasks using only internal knowledge?* 2) *Can full-sim search RL enable effective sim-to-real transfer with real web search during inference?*

First, we investigate whether an LLM can generate both queries and information based on the knowledge embedded in its parameters, effectively simulating querying external search engines. To this end, we assess the intrinsic search capabilities of LLMs on benchmarks that require web searching by prompting the model to simulate the search process within a single generation trajectory using multi-turn, tool-formatted outputs. Extensive sampling demonstrates that LLMs encode substantial world knowledge within their parameters, yielding high predictive `pass@k` scores that follow a scaling law. However, reliably extracting the optimal answer remains challenging, underscoring the gap between latent knowledge and actionable retrieval. To address this challenge and explore the potential of *full-sim* search RL for *sim-to-real* transfer, we study the potential of Self-Search Reinforcement Learning (SSRL) which enhances the self-search abilities of LLMs through format-based and rule-based rewards, enabling autonomous refinement of internal knowledge utilization without relying on external searches. Our experiments show that models trained with SSRL not only outperform previous search API-based RL baselines, such as Search-R1 and ZeroSearch, across various benchmarks, but also serve as cost-effective, implicit world knowledge provider, thus reducing hallucination, for search-driven question answering. Moreover, this approach reduces dependence on external search engines and opens new avenues for sim-to-real generalization, enabling skills acquired through self-search to transfer robustly to online settings with real web access.

In summary, our work demonstrates that LLMs hold significant potential as simulator of the web, a resource that can be leveraged for search-driven tasks without the need for external queries. By systematically quantifying and enhancing this self-search capability with SSRL, we pave the way for more autonomous and scalable LLM agents (Leike et al., 2018; Gao et al., 2025).

## 2 INFERENCE-TIME SCALING OF SELF-SEARCH

### 2.1 TASK FORMULATION

**Formulation of `pass@k`.** We consider the problem of answering information-seeking queries using only the internal knowledge of an LLM, without access to external retrieval tools such as web search engines or databases. We generate $K$ samples for problem $i$, and we calculate the number of accurate responses $C_i$. We compute $pass@k$ using the formula below:

$$\text{pass@k} = \frac{1}{\# \text{ of problems}} \sum_{i=1}^{\# \text{ of problems}} \left( 1 - \frac{\binom{K-C_i}{k}}{\binom{K}{k}} \right), \tag{1}$$

where correctness is defined according to the evaluation standard of the underlying benchmark (e.g., exact match, top-$k$ accuracy, or task-specific criteria). This setup allows us to estimate the intrinsic upper bound of the model's internalized search capabilities, independent of any external retriever.

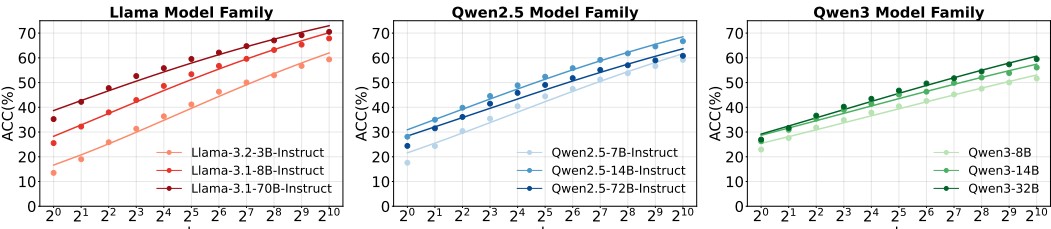

Figure 2: The scaling curves of repeated sampling averaged on six benchmarks within three model families (Qwen2.5, Llama, and Qwen3). It indicates predictive performance gains, where average MAE for different families is 1.42%, 1.45%, and 0.95%, respectively.

**Formulation of Scaling Law.** We present the formulation of the scaling law for test-time self-search. Following Brown et al. (2024), we define a function to simulate the correlation between the number of samples $K$ and the coverage $c$. We model the log of $c$ as a function of $k$ using:

$$\log c \approx ak^b, \tag{2}$$

where $a, b$ are fitted model parameters. We exponentiate each side to have a straightforward prediction of the coverage $c$. The final function can be presented as:

$$c \approx \exp(ak^b). \tag{3}$$

## 2.2 EXPERIMENTAL SETUP

**Benchmarks.** We evaluate across seven benchmarks spanning three categories of question-answering tasks: 1) General Question Answering, which tests factual knowledge retrieval using Natural Questions (Kwiatkowski et al., 2019) and TriviaQA (Joshi et al., 2017); 2) Multi-hop Question Answering, which requires reasoning across multiple pieces of information through HotpotQA (Yang et al., 2018), Musique (Trivedi et al., 2022), Bamboogle (Press et al., 2023), and 2Wiki-MultiHopQA (Ho et al., 2020); and 3) Vague Question Answering, which evaluates information synthesis from various vague restrictions using BrowseComp (Wei et al., 2025). This comprehensive evaluation framework captures capabilities ranging from direct knowledge retrieval to complex reasoning and information integration, providing a robust assessment of model performance across varied question-answering scenarios. Benchmark details are listed in Table 5.

**Models.** To ensure comprehensive evaluation of the effects of repeated sampling, we conduct experiments across three model families: Qwen2.5 (Qwen et al., 2025), Llama3 (including Llama-3.1 and Llama-3.2) (Grattafiori et al., 2024), and Qwen3 (Yang et al., 2025a). We test models spanning a wide range of parameter scales from 0.6B to 72B. To ensure a fair comparison across all experiments, we maintain consistent sampling parameters with temperature set to 0.7, `top-k` to -1, `top-p` to 0.95, and max token to 8192. The prompt used is listed in Section 3.2.

## 2.3 PERFORMANCE EVALUATION

**Predictive Performance Improves with Sample Size.** As shown in Figure 2 and Figure 3, we observe consistent and predictive performance improvements across all benchmarks as the number of samples increases. Notably, on Bamboogle, Llama-3.1-8B-Instruct achieves 87.2% accuracy for `pass@1024`, a **150%** improvement over `pass@1` performance. These substantial gains are evident across all three model families (Qwen2.5, Llama, and Qwen3), with the Llama series showing particularly pronounced benefits. Figure 3 shows performance on BrowseComp, a benchmark characterized by difficult search requirements but straightforward verification. While GPT-4o with search achieves only 1.9% and o1 scores 10%, Self-Search yields surprising results: Qwen2.5-14B-Instruct and Llama-3.1-8B-Instruct surpass o1's performance when given sufficient samples. This finding suggests that LLMs possess substantial internal knowledge that can be effectively leveraged through repeated sampling, even in the absence of external information sources. Analysis of the upper bound further highlights the strong potential of LLMs to provide information in response to given search actions. More details are provided in Appendix A.2.

**Llama Outperforms Qwen, Contrary to Prior Reasoning Tasks.** Previous works (Gandhi et al., 2025; Liu et al., 2025b; Wang et al., 2025a) have shown that Qwen models (including Qwen2.5 and Qwen3) possess stronger priors in mathematical reasoning and achieve greater improvements than Llama models in RL settings. However, our findings indicate that Llama models outperform Qwen models in the Self-Search setting with respect to priors for world knowledge, as demonstrated in Figure 2 and Figure 3. This observation suggests that self-search ability and reasoning priors are not strongly correlated. We will further explore the utilization of knowledge and reasoning in Appendix A.3.

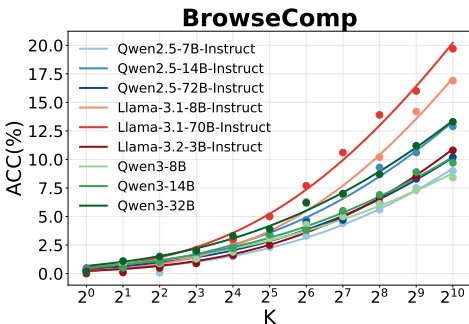

Figure 3: TTS on BrowseComp leads to consistent performance gains within all models. It indicates predictive performance gains, with average MAE for the LLaMA, Qwen 2.5, and Qwen 3 families at 0.34 %, 0.22 %, and 0.26 %, respectively.

**Performance Gap Narrows Between Large and Small Models with More Sampling.** Remarkably, our results demonstrate that smaller models can achieve performance comparable to models with nearly $10\times$ more parameters by repeated sampling, as measured by `pass@k`. For example, on TriviaQA with 1024 samples, Llama-3.1-8B-Instruct achieves a score of 81.2%, while Llama-3.1-70B-Instruct achieves 81.4%, a negligible difference despite the substantial gap in model size. This finding is consistent with previous studies (Snell et al., 2024; Liu et al., 2025a), indicating the potential of small language models.

### 2.4 Majority Voting vs. Pass@k

In the above experiment, we found that LLMs exhibit a high performance ceiling in search and question-answering tasks. However, it remains challenging to identify the correct answer from a set of candidate responses, despite the correct answer being present, when the ground truth is unknown (Brown et al., 2024). This suggests that repeated sampling represents the upper limit of Test-Time Scaling (TTS), and further evaluation of alternative TTS strategies is necessary.

Majority voting is widely thought of as a simple but effective method to integrate with Test-time Scaling (Zuo et al., 2025). To investigate whether the performance transfers to knowledge-intensive tasks, we employ the `maj@k` metric to have an overview.

In essence, `maj@k` evaluates to 1 when the most frequently occurring answer among $K$ samples matches the ground truth, and 0 otherwise. The instruction for the majority voting experiments is detailed in Appendix 3.2. We show the results in Figure 9. Our experiments reveal that even as we increase the number of responses $k$ for majority voting, we observe only marginal performance improvements. This limited scaling behavior suggests that naive majority voting may be insufficient for search tasks, where incorrect answers might consistently appear across multiple samples. These findings indicate that LLMs have the potential to become world models, but the world knowledge presented is vague, and how to provide precise knowledge is still a challenging task.

## 3 SSRL: Self-Search Reinforcement Learning

### 3.1 Task Definition

We formulate the RL objective for LLM-based search agent utilizing external search engines as:

$$\max_{\pi_\theta} \mathbb{E}_{x\sim D, y\sim\pi_\theta(\cdot|x;R)}[r_\phi(x,y)] - \beta\mathbb{D}_{KL}[\pi_\theta(y|x;R)||\pi_{\text{ref}}(y|x;R)], \quad (4)$$

where $\pi_\theta$ denotes the policy model, $\pi_{\text{ref}}$ represents the reference model, $r_\phi$ is the reward function, $R$ represents retrieved information, and $\mathbb{D}_{KL}$ denotes the KL divergence regularization term with coefficient $\beta$. In our approach, since the model auto-regressively retrieves knowledge from its internal parameters rather than external sources, the retrieved information $R$ follows the same distribution as $\pi_\theta$. This Self-Search mechanism allows us to simplify the objective function to:

$$\max_{\pi_\theta} \mathbb{E}_{x\sim D, y\sim\pi_\theta(\cdot|x)}[r_\phi(x,y)] - \beta\mathbb{D}_{KL}[\pi_\theta(y|x)||\pi_{\text{ref}}(y|x)], \quad (5)$$

where $\pi_\theta$ simultaneously functions as both the reasoning policy model and the internal search engine. We primarily leverage GRPO (Shao et al., 2024) as our training algorithm, while also experimenting with other RL algorithms, e.g., PPO (Schulman et al., 2017), DAPO (Yu et al., 2025).

## 3.2 PROMPT DESIGN

Following Jin et al. (2025a), we use an unbiased instruction without any hints for reflection. The instruction just teaches LLMs to think step by step. The prompt template is shown in Table 1.

Table 1: Prompt template. The question is appended at the end during training and inference.

---

**Prompt Template**

---

Answer the given question. You must conduct reasoning inside `<think>` and `</think>` first every time you get new information. After reasoning, if you find you lack some knowledge, you can call a search engine by `<search>` query `</search>`, and you should return the top searched results between `<information>` and `</information>`. You can search as many times as you want. For multi-hop QA, you can break it down into pieces and search one by one. If you find no further external knowledge needed, you can directly provide the answer inside `<answer>` and `</answer>` without detailed illustrations. For example, `<answer>` Beijing `</answer>`. Question:

---

Our iterative reasoning framework follows a structured process where the model first expresses its initial thoughts within `<think>`...`</think>` tags. When the model identifies missing information necessary for solving the problem, it formulates search queries within `<search>`...`</search>` tags. The model then auto-regressively generates relevant information to address these queries, which is incorporated within `<information>`...`</information>` tags. This cycle of thinking, searching, and information gathering continues iteratively until the model arrives at a final answer. While this approach shares similarities with traditional multi-turn search systems, it fundamentally differs in its implementation: rather than conducting genuine iterative interactions with external systems, our method employs a Chain-of-Thought (Wei et al., 2023) process where the language model auto-regressively generates the entire reasoning trajectory in a single forward pass, including thoughts, search queries, and retrieved information. This design enables efficient self-contained search while maintaining the structured exploration benefits of iterative search processes.

## 3.3 TRAINING METHODOLOGY

**Information Token Mask** Previous research (Jin et al., 2025b; Sun et al., 2025) demonstrates that masking information tokens from external search engines helps stabilize training and improve performance. However, in Self-Search, the retrieved information originates from the model's own generation process rather than external sources, raising questions about whether information masking remains beneficial in this context. To investigate this, we conduct comparative experiments under two conditions: training with complete reasoning trajectories versus training with information-masked trajectories. For implementation, we extract all the tokens embraced by `<information>` and `</information>` and mask them for loss calculation. Our results reveal that information masking continues to enhance performance even when the information is self-generated by the model. We show our detailed experiments in Appendix B.3.1.

**Reward Modeling** Following DeepSeek-AI (2025); Yu et al. (2025), we employ a composite reward function with two signals: format reward and outcome reward. We directly use the accuracy of the model's final prediction as the outcome reward, computed using the following rule:

$$R(\hat{y}, y) = \begin{cases} 1, & \texttt{is\_equivalent}(\hat{y}, y) \\ -1, & \text{otherwise} \end{cases} \tag{6}$$

where $y$ is the ground-truth answer and $\hat{y}$ is the predicted answer.

Since our iterative search process requires models to decompose complex questions into manageable sub-problems, with each iteration focusing on searching for specific information and incrementally building toward the final answer, maintaining a structured output format is crucial for effective reasoning. To address this requirement, we introduce a format reward that ensures adherence to the prescribed reasoning structure, detailed in Appendix C. This format reward guides the model

Table 2: Main results of our trained models on the six benchmarks measured by EM. The column **Search Engine** refers to the external search engine used in the training stage and the evaluation stage. We use $\emptyset$ to denote that the baseline does not undergo the stage and "-" to denote using internal knowledge. We use $\infty$ to denote the Simulation LLM and W to denote Wikipedia for simplification. We use G to denote Google. The largest score of each model is denoted using **bold**.

| Model | Search Engine | General QA | | Multi-Hop QA | | | | Avg |
|---|---|---|---|---|---|---|---|---|
| | | NQ | TQ | HotpotQA | Musique | 2Wiki | Bamboogle | |
| LLaMA-3.2-3B | | | | | | | | |
| Direct Answer | $\emptyset$/- | 16.2 | 29.6 | 12.6 | 2.0 | 9.2 | 8.0 | 12.9 |
| CoT | $\emptyset$/- | 26.2 | 44.4 | 16.0 | 5.8 | 10.2 | 21.6 | 20.7 |
| RAG | $\emptyset$/G | 30.0 | 57.6 | 23.4 | 9.6 | 17.6 | 11.2 | 24.9 |
| Search-o1 | $\emptyset$/G | 24.2 | 48.4 | 19.4 | 6.0 | 17.4 | 32.0 | 24.6 |
| R1-Base | -/- | 28.4 | 44.2 | 22.8 | 7.0 | 28.4 | 11.1 | 23.7 |
| R1-Instruct | -/- | 35.0 | 52.2 | 21.6 | 11.4 | 17.8 | 20.8 | 26.5 |
| Search-R1-Base | W/G | 41.2 | 60.0 | 29.6 | 13.6 | **31.6** | 19.4 | 32.6 |
| Search-R1-Instruct | W/G | 37.6 | 53.6 | 21.0 | 8.8 | 20.4 | 27.8 | 28.2 |
| ZeroSearch-Base | $\infty$/G | 43.4 | **63.8** | **32.2** | 13.8 | 35.6 | 15.3 | 34.0 |
| ZeroSearch-Instruct | $\infty$/G | 40.2 | 58.0 | 22.8 | 10.4 | 21.4 | 18.1 | 28.5 |
| SELF-SEARCH-BASE | -/- | 35.0 | 45.8 | 28.2 | 14.2 | 29.6 | 30.2 | 30.5 |
| SELF-SEARCH-INSTRUCT | -/- | **43.8** | 58.4 | 25.0 | **14.2** | **31.6** | **38.4** | **35.2** |
| LLaMA-3.1-8B | | | | | | | | |
| Direct Answer | $\emptyset$/- | 21.2 | 52.8 | 21.0 | 3.2 | 8.0 | 23.8 | 21.7 |
| CoT | $\emptyset$/- | 23.0 | 46.6 | 18.8 | 8.8 | 17.6 | 35.2 | 25.0 |
| RAG | $\emptyset$/G | 40.8 | 62.8 | 37.0 | 22.4 | 34.0 | 38.4 | 39.2 |
| Search-o1 | $\emptyset$/G | 26.8 | 37.2 | 21.0 | 9.2 | 23.6 | 25.6 | 23.9 |
| R1-Base | -/- | 21.0 | 48.8 | 23.0 | 5.4 | 28.0 | 5.6 | 22.0 |
| R1-Instruct | -/- | 39.2 | 59.8 | 30.4 | 18.2 | 36.8 | 47.2 | 38.6 |
| Search-R1-Base | W/G | 41.0 | 62.6 | **40.0** | **25.0** | **37.8** | 36.1 | 40.4 |
| Search-R1-Instruct | W/G | 39.6 | 59.6 | 36.8 | 19.6 | 34.8 | 31.9 | 37.1 |
| ZeroSearch-Base | $\infty$/G | 38.2 | 52.4 | 26.0 | 9.6 | 28.4 | 12.5 | 27.9 |
| ZeroSearch-Instruct | $\infty$/G | **48.2** | **68.2** | 36.6 | 19.6 | 36.2 | 40.3 | 41.5 |
| SELF-SEARCH-BASE | -/- | 41.0 | 49.6 | 30.0 | 18.4 | 34.4 | 32.8 | 34.4 |
| SELF-SEARCH-INSTRUCT | -/- | 48.0 | 62.6 | 34.4 | 24.2 | 35.2 | **54.4** | **43.1** |

to produce well-organized, multi-step reasoning trajectories. We also provide an ablation study in Appendix B.3.2, showing the effectiveness of format reward.

The final reward combines both components as:

$$
r_\phi(y_i, y) = \begin{cases} 1 & \text{if } \texttt{is\_equivalent}(\hat{y}, y) \wedge f_{\text{format}}(y) = \text{True}, \\ 1 - \lambda_f & \text{if } \texttt{is\_equivalent}(\hat{y}, y) \wedge f_{\text{format}}(y) = \text{False}, \\ \lambda_f & \text{if } !\texttt{is\_equivalent}(\hat{y}, y) \wedge f_{\text{format}}(y) = \text{True}, \\ 0 & \text{if } !\texttt{is\_equivalent}(\hat{y}, y) \wedge f_{\text{format}}(y) = \text{False}, \end{cases}
\tag{7}
$$

where $\lambda_f = 0.1$, prioritizing correctness yet maintaining structured format, as (Wang et al., 2025b).

## 3.4 EXPERIMENTAL SETUP

**Benchmarks** We conduct evaluation across six of the benchmarks described in Section 2.2. We exclude BrowseComp from evaluation due to its exceptional difficulty and limited availability of training data. To ensure fair comparison with existing baselines, we adopt the same validation sets used by Sun et al. (2025). Our evaluation employs EM, where a prediction is considered correct only when it matches the ground truth answer precisely.

**Baselines** To evaluate the effectiveness of Self-Search, we compare our model with the following methods: **Vanilla Prompt Methods:** It includes Direct Prompt and CoT; **RAG-based Methods:** This category includes standard RAG and Search-o1 (Li et al., 2025c); **RL-based Methods:** This category includes R1, Search-R1 (Jin et al., 2025b), and ZeroSearch (Sun et al., 2025). We conduct offline evaluations of our models while enabling online testing for baseline methods where applicable. To ensure fair comparison in online settings, we limit the number of retrieved passages to 3 across all RAG-based approaches. For vanilla prompt methods, we employ instruction-tuned models as they demonstrate superior prompt-following capabilities. The implementation details of baselines are listed in Appendix B.2.1.

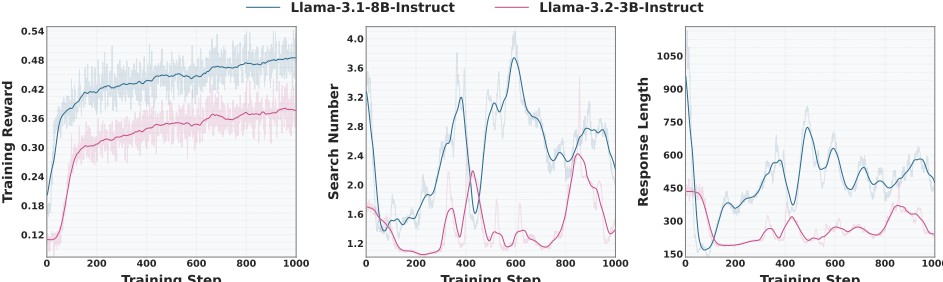

Figure 4: The training curves of Llama-3.2-3B-Instruct and Llama-3.1-8B-Instruct. The three figures are the training reward, the response length, and the number of searches included.

Table 3: Performance of Sim2Real Search Generalization. The largest score is denoted using **bold**. The second largest score is denoted using underline.

| Model | General QA | | Multi-Hop QA | | | | Avg |
|---|---|---|---|---|---|---|---|
| | NQ | TQ | HotpotQA | Musique | 2Wiki | Bamboogle | |
| LLaMA-3.2-3B-Instruct | | | | | | | |
| Zero-shot CoT | 26.2 | 44.4 | 16.0 | 5.8 | 10.2 | 21.6 | 20.7 |
| SSRL | 43.8 | 58.4 | 25.0 | 14.2 | 31.6 | 38.4 | 35.2 |
| Sim2Real ($K$=1) | 44.4 | **63.4** | 34.8 | 17.2 | 37.8 | 42.4 | 40.0 |
| Sim2Real ($K$=3) | **44.8** | 63.0 | **35.4** | 19.4 | 41.8 | **47.2** | **41.9** |
| Sim2Real (All) | 44.0 | 61.6 | 35.2 | **20.8** | **42.8** | 46.4 | 41.8 |
| LLaMA-3.1-8B-Instruct | | | | | | | |
| Zero-shot CoT | 23.0 | 46.6 | 18.8 | 8.8 | 17.6 | 35.2 | 25.0 |
| SSRL | **48.0** | **62.6** | 34.4 | 24.2 | 35.2 | **54.4** | **43.1** |
| Sim2Real ($K$=1) | 39.4 | 55.8 | 34.0 | **26.8** | **39.8** | 53.6 | 41.6 |
| Sim2Real ($K$=3) | 33.2 | 50.6 | 29.7 | 23.4 | 39.2 | 36.6 | 35.5 |
| Sim2Real (All) | 39.6 | 54.6 | **34.6** | 25.0 | 36.8 | 50.4 | 40.2 |
| Qwen2.5-3B-Instruct | | | | | | | |
| Zero-shot CoT | 15.0 | 33.6 | 16.2 | 3.6 | 18.0 | 12.8 | 14.7 |
| SSRL | 23.6 | 41.0 | 22.4 | 10.4 | 26.0 | **32.8** | 26.0 |
| Sim2Real ($K$=1) | 35.2 | 44.0 | 22.0 | 14.8 | 36.6 | 26.4 | 29.8 |
| Sim2Real ($K$=3) | **37.8** | **51.6** | 26.4 | 22.4 | **36.8** | 21.6 | 32.8 |
| Sim2Real (All) | **37.8** | 51.4 | **27.4** | **22.4** | 36.4 | 22.4 | **33.0** |
| Qwen2.5-7B-Instruct | | | | | | | |
| Zero-shot CoT | 12.8 | 35.6 | 16.2 | 6.6 | 22.6 | 24.0 | 17.4 |
| SSRL | 31.4 | 44.4 | 26.0 | 11.8 | 31.0 | 36.8 | 30.2 |
| Sim2Real ($K$=1) | 38.4 | 58.0 | 35.6 | 18.4 | 36.0 | 41.6 | 38.0 |
| Sim2Real ($K$=3) | **43.8** | 64.4 | 42.0 | **29.4** | **53.4** | **54.5** | **47.9** |
| Sim2Real (All) | 41.8 | **65.0** | **43.2** | 28.6 | 50.4 | 52.0 | 46.8 |

**Training Setups** We conduct our RL experiments primarily on the Llama model family, specifically Llama-3.2-3B (Base/Instruct) and Llama-3.1-8B (Base/Instruct), selected based on their demonstrated effectiveness under repeated sampling conditions. We use the combination of the training dataset of NQ and HotpotQA, as in previous work (Jin et al., 2025b), to ensure a mix of general QA and multi-hop QA. Our training framework primarily utilizes GRPO as the default algorithm, while also experimenting with alternative approaches, including PPO and REINFORCE++, to validate the robustness of our findings. All training is conducted on a single node equipped with 8 NVIDIA A800 GPUs. For GRPO, the training configuration includes a batch size of 256, a learning rate of 1e-6, and 62 warmup steps across all experiments. The max response length is 4096 across all models in our experiments. For policy optimization, we set the temperature to 1.0 and generate 5 rollouts per prompt. We apply a KL divergence coefficient of 0.001. We train each model for 5 epochs and select the checkpoint with the highest average validation accuracy for final evaluation, ensuring optimal performance while preventing overfitting. For all the evaluation, we set the temperature to 0.0. We conduct a group size ablation for GRPO in Appendix B.3.9 and a temperature ablation in Appendix B.3.10. The implementation details of other algorithms are listed in Appendix B.2.2, and the results are listed in Appendix B.3.5.

Table 4: We display key concepts discussed in this paper. The terminology we mentioned above is the approach of search used during training and inference.

| Terminology | Explanation | Example |
|---|---|---|
| Full-Real Search | Search external real engines like RAG or Google. | Search-R1 (Jin et al., 2025b) |
| Semi-Real Search | Search external simulated engines like LLMs. | ZeroSearch (Sun et al., 2025) |
| Full-Sim Search | Search internal engines, e.g., implicitly retrieving information from embedded knowledge. | SELF-SEARCH |
| Sim2Real Search | Train with Full-Sim Search but inference with external real engines, such as Google Search or Bing. | SELF-SEARCH |

## 3.5 PERFORMANCE EVALUATION

### 3.5.1 SELF-SEARCH RL

We present the main experimental results in Table 2 and show the case studies in Appendix (Table 31 and Table 32). We also experiment on the Qwen series, and the results of Qwen2.5 and Qwen3 are listed in Appendix B.3.6 and Appendix B.3.7. The results reveals several key insights:

**SSRL achieves superior performance.** Our results demonstrate that models trained with auto-regressive internal retrieval consistently outperform those relying on external search engines, whether using other LLMs or Google Search. We also observe a better performance compared with R1-like models, which are trained with the naive CoT prompt. These findings suggest that through well-designed instruction and reward, language models can effectively function as both reasoners and knowledge retrievers simultaneously, successfully extracting relevant information from their internal parametric knowledge without external dependencies. We further show the importance of the on-policy information simulator in Appendix B.3.3. Besides, we conduct an empirical study on the quality of the generated information and the hallucination analysis in Appendix B.3.4.

**Instruction models more effectively utilize internal knowledge.** When trained on identical data for the same duration, instruction-tuned models achieve significantly better performance than their base counterparts, suggesting that additional knowledge operations may be incorporated during supervised fine-tuning. However, this advantage appears to be context-dependent: while instruction-tuned models excel at leveraging internal knowledge, base models demonstrate superior performance when external information sources are available. This finding implies that different optimization strategies are required for internal versus external knowledge utilization.

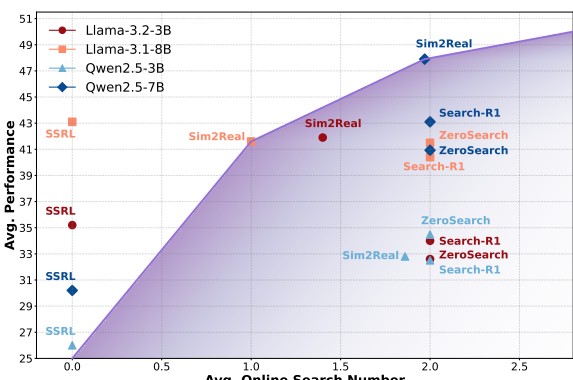

Figure 5: Pareto frontier illustrating the trade-off between performance and the number of real searches across different models. The Sim2Real models are evaluated using the maximum score within Sim2Real ($K = 1$), Sim2Real ($K = 3$) for fair comparison.

**Larger models show better self-search performance.** Figure 4 presents the training curves. We observe steady growth in training reward throughout the process. During the early training stage, both response length and search count decrease as the models adapt to the format reward constraints. In later stages, Llama-3.1-8B-Instruct develops more sophisticated strategies, learning to decompose questions and employ self-reflection to enhance performance, thus yielding a better performance on our benchmarks.

**SSRL is more efficient and robust.** Figure 10 presents the training curves for ZeroSearch and SSRL. Compared to ZeroSearch, SSRL demonstrates substantially improved training efficiency, achieving a 5.53× reduction in training time. Additionally, SSRL exhibits steady reward growth throughout training without collapse, indicating robust performance. Although SSRL shows relatively lower training rewards than ZeroSearch during early training stages, its superior efficiency and robustness compensate for this initial disadvantage.

### 3.5.2 SIM2REAL GENERALIZATION

Although SSRL achieves strong results on static benchmarks, the inherent knowledge within these models remains fixed, which limits their applicability to real-world scenarios. In this work, we investigate whether SSRL can generalize to real-time search settings. Since our trained model follows the exact format specifications of Search-R1 (Jin et al., 2025b), we can seamlessly integrate real search capabilities. We refer to this setting as *sim-to-real* generalization, following terminology from prior work in Robotics RL (Kaspar et al., 2020; Da et al., 2025).

We replace model-generated information with results of actual searches from Google Search or local corpora, substituting up to $K$ self-generated responses, where $K$ represents the maximum turns used by Jin et al. (2025b). To ensure compatibility, we post-process the retrieved information using rule-based modifications that remove patterns that hinder the reasoning of models. Table 3 and Figure 5 present our experimental results. We also experiment on SimpleQA (Wei et al., 2024) in Appendix B.3.11. We can observe that performance consistently improves with an increasing number of maximum turns across all models except Llama-3.1-8B-Instruct. Furthermore, compared to Search-R1 and ZeroSearch baselines, SSRL-based models generally achieve superior performance with less online searching across various benchmarks under Sim2Real settings. These findings demonstrate that search agents trained exclusively on internal knowledge can effectively leverage external knowledge sources when format alignment is maintained, thereby reducing training costs and improving efficiency. Case study is in Table 33. We further show the trial of entropy-guided Sim2Real search in Appendix B.3.13 to control the search cost.

### 3.5.3 TEST-TIME RL

Considering unsupervised RL algorithms, e.g., TTRL, (Zuo et al., 2025), show great potential in math and code generation, we are curious about its generalization to SELF-SEARCH. We conduct experiments on the Llama series, using the dataset consisting of NQ, TQ, HotpotQA, MusiQue, Bamboogle, 2WikiMultiHopQA, and BrowseComp. We also experiment with Sim2Real Generalization on TTRL + SSRL trained models. The result is listed in Appendix B.4, where we achieve better performance on BrowseComp compared with WebSailor (Li et al., 2025b).

## 4 RELATED WORK

**Reinforcement Learning with Search Engines** RL has proven effective in boosting LLM reasoning via process or outcome rewards, enabling strong performance on math and code through self-reflection and exploration (DeepSeek-AI, 2025; Cui et al., 2025; OpenAI, 2024). Recent works extend RL to LLM-based search agents: Search-R1 trains iterative corpus search with retrievers (Jin et al., 2025a). ReSearch leverages outcome rewards to improve information seeking (Chen et al., 2025). However, they rely on static corpora, lacking the complexity of real-world search. To bridge this gap, online-search RL couples LLMs with web engines (Google, Bing) (Zheng et al., 2025), though API-heavy algorithms such as GRPO incur high costs (Shao et al., 2024). A cost-effective alternative, ZeroSearch, simulates search environments using LLMs themselves, reducing overhead while maintaining performance (Sun et al., 2025). Yet, the potential of LLMs as world models for RL-based search remains underexplored, and their upper limits in agentic search are still unknown.

**Large Language Models as Search Engines** With the rise of LLMs, generative search has emerged as a new paradigm, offering flexible, multi-grained information through generation rather than traditional retrieval (Li et al., 2024b; 2025d). Existing work mainly explores two directions: generative retrieval, which directly generates document identifiers as implicit knowledge bases (Tay et al., 2022; Wang et al., 2022; Li et al., 2024c; Long et al., 2024); and reliable response generation, where LLMs summarize retrieved items (e.g., papers, web pages) into user-centric answers (Gao

et al., 2023; Qin et al., 2023; Shen et al., 2023). These approaches overcome rigid document granularity and relevance matching, enabling more flexible and creative applications (Li et al., 2024a; Ding et al., 2025). However, their use as textual world models in agentic RL remains underexplored.

**Inference-time Scaling of LLMs and Agents** Repeated sampling generates multiple candidate outputs from the same prompt, with studies showing that correct-answer coverage improves as sample size increases (Brown et al., 2024; Yue et al., 2025; Li et al., 2025a). Beyond raw sampling, methods such as Best-of-N (Liu et al., 2025a; Qiu et al., 2024) and majority voting (Zuo et al., 2025) leverage verification to further enhance performance. However, their application in search contexts remains underexplored. Meanwhile, TTS has been extended from reasoning to agentic systems. Recent work demonstrates the effectiveness of parallel sampling, reflective revision, and diversified rollouts in language agents (Zhu et al., 2025), while interaction-based scaling enables adaptive exploration and backtracking for web agents (Shen et al., 2025). Further, approaches such as GUI Test-time Scaling (Yang et al., 2025c) and Multi-Agent Verification (Lifshitz et al., 2025) highlight the potential of both compute allocation and multi-agent verification to improve robustness across diverse environments.

## 5 CONCLUSION

In conclusion, our study establishes that LLMs possess untapped capacity as implicit world models for search-driven tasks, often containing the necessary knowledge to answer complex queries internally. While reliably extracting this knowledge remains difficult, our proposed Self-Search Reinforcement Learning (SSRL) method significantly enhances self-search abilities, outperforming search API-based baselines and enabling robust sim-to-real transfer. These findings suggest a promising path toward more autonomous and scalable LLM agents that can operate effectively without reliance on external search engines.

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

Table 5: Benchmark concerning search. Most benchmarks are constructed manually, except 2Wiki-MultiHopQA and MuSiQue. Most of the benchmarks are designed for QA initially.

| Knowledge Type | Benchmark | Time | Construction | Targeted Task | Source |
|---|---|---|---|---|---|
| **Factual** | TriviaQA | 2017 | Manual | General QA | Wikipedia + Web |
| | Natural Questions | 2019 | Manual | General QA | Wikipedia |
| | SimpleQA | 2024 | Manual | Factual QA | General knowledge |
| **Reason** | HotpotQA | 2018 | Manual | Multi-hop QA | Wikipedia |
| | 2WikiMultiHopQA | 2020 | Semi-automated | Multi-hop QA | Wikipedia + Wikidata |
| | BamBoogle | 2022 | Manual | Multi-hop QA | Wikipedia |
| | MuSiQue | 2022 | Automated | Multi-hop QA | Wikipedia |
| **Web browsing** | BrowseComp | 2025 | Manual | Search and Browse | Open Web |

## A  INFERENCE-TIME SCALING OF SELF-SEARCH

### A.1  PROMPTS

#### A.1.1  INSTRUCTIONS FOR LLM PROVIDING INFORMATION

We use the instruction in Table 6 when querying LLM to provide information.

Table 6: Instruction for LLM providing information.

---

Given a query, you need to imitate the style of the following demos and generate five useful documents for the query.
`[EXAMPLE]`
You should generate documents that can help the user find the answer. Each document should contain about 30 words. You must directly output the English documents and not output any other texts.
Query: query
Useful Output:

---

### A.2  DETAILED RESULTS

We introduce the results of repeated sampling of seven benchmarks, across 16 models, in Figure 6. We also list the simulated parameters for each model in Table 7 and comparison of actual vs. fitted values, residuals, and relative errors for every model across different $k$ values in Table 8, 9, 10

Table 7: Fitting performance metrics for Llama and Qwen series models (RMSE and MAE are in percentage)

| Model | $a$ | $b$ | $R^2$ | RMSE | MAE |
|---|---|---|---|---|---|
| Llama-3.2-3B-Instruct | $-1.793$ | $-0.191$ | 0.986 | 1.745% | 1.583% |
| Llama-3.1-8B-Instruct | $-1.263$ | $-0.183$ | 0.987 | 1.541% | 1.267% |
| Llama-3.1-70B-Instruct | $-0.950$ | $-0.159$ | 0.976 | 1.688% | 1.415% |
| Qwen3-8B | $-1.370$ | $-0.111$ | 0.984 | 1.115% | 0.906% |
| Qwen3-14B | $-1.249$ | $-0.117$ | 0.984 | 1.184% | 0.987% |
| Qwen3-32B | $-1.232$ | $-0.130$ | 0.989 | 1.073% | 0.949% |
| Qwen2.5-7B-Instruct | $-1.533$ | $-0.167$ | 0.978 | 1.932% | 1.674% |
| Qwen2.5-14B-Instruct | $-1.174$ | $-0.163$ | 0.989 | 1.265% | 1.029% |
| Qwen2.5-72B-Instruct | $-1.259$ | $-0.148$ | 0.970 | 1.984% | 1.660% |

### A.3  IS MORE REASONING ALWAYS BETTER?

**Experimental Setup.**  As discussed in Section 2.3, we observe inconsistent results on the agentic search benchmark of reasoning and instruction models, e.g., Qwen3 vs Qwen2.5 and Llama. In

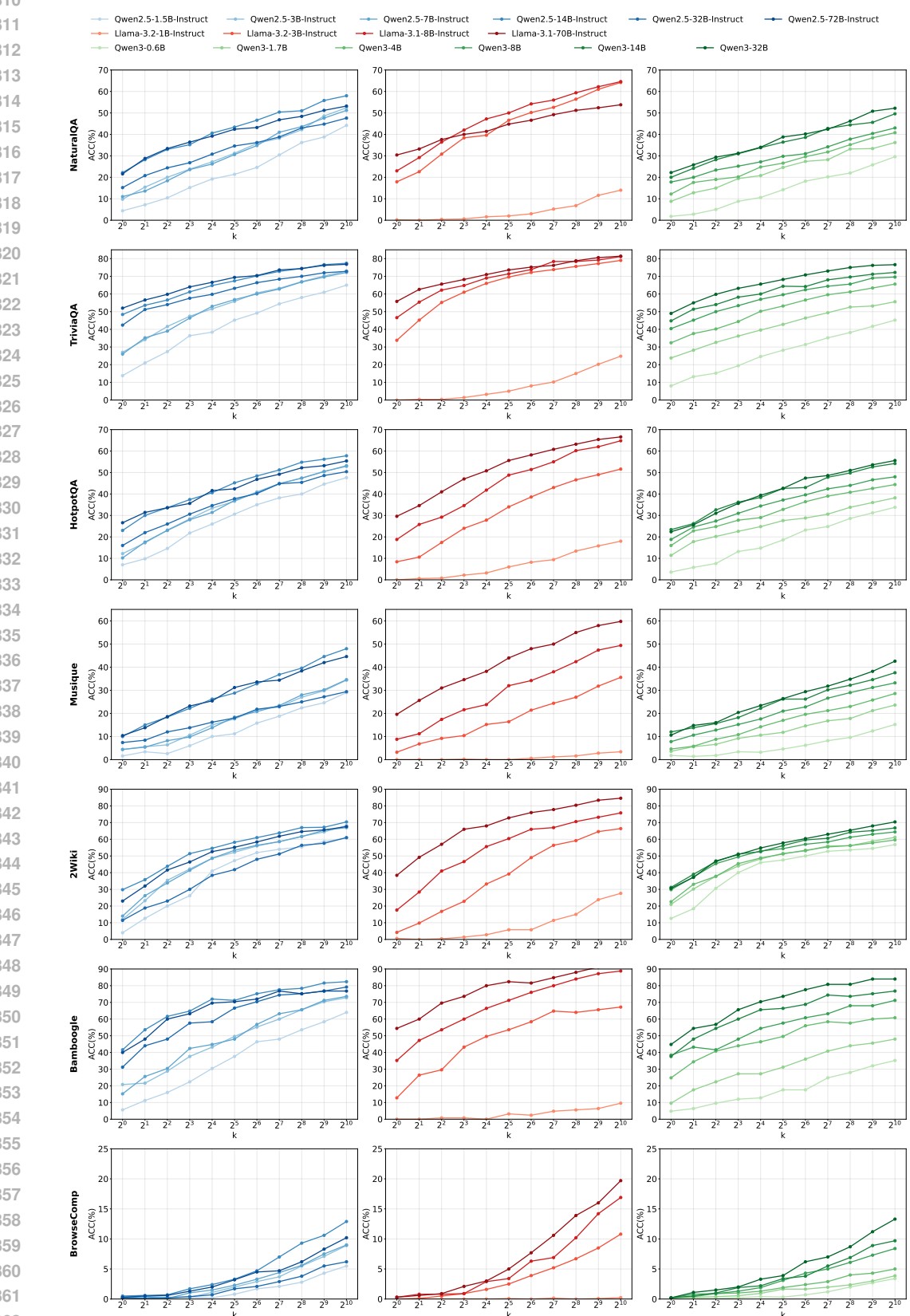

Figure 6: The results of repeated sampling of seven benchmarks

Table 8: Comparison of actual vs. fitted values, residuals, and relative errors for three Llama models across different $K$ values.

| K | Llama-3.2-3B-Instruct | | | | Llama-3.1-8B-Instruct | | | | Llama-3.1-70B-Instruct | | | |
|---|---|---|---|---|---|---|---|---|---|---|---|---|
| | Actual | Fitted | Residual | Rel. Error (%) | Actual | Fitted | Residual | Rel. Error (%) | Actual | Fitted | Residual | Rel. Error (%) |
| $2^0$ | 13.50 | 16.64 | -3.14 | 23.26 | 25.53 | 28.27 | -2.74 | 10.73 | 35.23 | 38.69 | -3.45 | 9.80 |
| $2^1$ | 19.00 | 20.78 | -1.78 | 9.35 | 32.20 | 32.87 | -0.67 | 2.09 | 42.20 | 42.72 | -0.52 | 1.24 |
| $2^2$ | 25.88 | 25.24 | 0.64 | 2.49 | 37.97 | 37.54 | 0.43 | 1.12 | 47.77 | 46.69 | 1.07 | 2.24 |
| $2^3$ | 31.32 | 29.92 | 1.40 | 4.45 | 42.93 | 42.20 | 0.74 | 1.71 | 52.63 | 50.56 | 2.07 | 3.93 |
| $2^4$ | 36.36 | 34.74 | 1.62 | 4.44 | 48.63 | 46.78 | 1.86 | 3.82 | 55.83 | 54.30 | 1.54 | 2.75 |
| $2^5$ | 41.16 | 39.60 | 1.56 | 3.79 | 53.37 | 51.22 | 2.15 | 4.03 | 59.53 | 57.88 | 1.66 | 2.78 |
| $2^6$ | 46.28 | 44.41 | 1.87 | 4.04 | 56.73 | 55.47 | 1.26 | 2.22 | 62.13 | 61.28 | 0.85 | 1.37 |
| $2^7$ | 50.04 | 49.10 | 0.94 | 1.87 | 59.60 | 59.52 | 0.08 | 0.14 | 64.73 | 64.50 | 0.24 | 0.36 |
| $2^8$ | 52.96 | 53.62 | -0.66 | 1.25 | 63.17 | 63.32 | -0.15 | 0.24 | 67.03 | 67.52 | -0.49 | 0.73 |
| $2^9$ | 56.72 | 57.92 | -1.20 | 2.12 | 65.33 | 66.87 | -1.53 | 2.35 | 69.17 | 70.35 | -1.18 | 1.71 |
| $2^{10}$ | 59.36 | 61.97 | -2.61 | 4.40 | 67.83 | 70.16 | -2.33 | 3.43 | 70.50 | 72.98 | -2.48 | 3.52 |

Table 9: Comparison of actual vs. fitted values, residuals, and relative errors for Qwen3-8B, Qwen3-14B, and Qwen2.5-7B-Instruct.

| K | Qwen3-8B | | | | Qwen3-14B | | | | Qwen2.5-7B-Instruct | | | |
|---|---|---|---|---|---|---|---|---|---|---|---|---|
| | Actual | Fitted | Residual | Rel. Error (%) | Actual | Fitted | Residual | Rel. Error (%) | Actual | Fitted | Residual | Rel. Error (%) |
| $2^0$ | 22.92 | 25.42 | -2.50 | 10.90 | 26.28 | 28.69 | -2.41 | 9.15 | 17.60 | 21.58 | -3.98 | 22.63 |
| $2^1$ | 27.56 | 28.14 | -0.58 | 2.09 | 30.92 | 31.62 | -0.70 | 2.26 | 24.33 | 25.51 | -1.18 | 4.83 |
| $2^2$ | 31.80 | 30.91 | 0.89 | 2.79 | 35.40 | 34.59 | 0.81 | 2.29 | 30.40 | 29.61 | 0.79 | 2.61 |
| $2^3$ | 34.84 | 33.73 | 1.11 | 3.20 | 38.96 | 37.57 | 1.39 | 3.56 | 35.40 | 33.81 | 1.59 | 4.50 |
| $2^4$ | 37.84 | 36.56 | 1.28 | 3.39 | 41.40 | 40.55 | 0.85 | 2.04 | 40.43 | 38.05 | 2.38 | 5.89 |
| $2^5$ | 40.40 | 39.39 | 1.01 | 2.50 | 45.16 | 43.51 | 1.65 | 3.65 | 44.43 | 42.28 | 2.16 | 4.85 |
| $2^6$ | 42.56 | 42.21 | 0.35 | 0.82 | 46.32 | 46.43 | -0.11 | 0.23 | 47.43 | 46.44 | 1.00 | 2.10 |
| $2^7$ | 45.20 | 45.00 | 0.20 | 0.44 | 49.88 | 49.29 | 0.59 | 1.18 | 51.23 | 50.49 | 0.74 | 1.45 |
| $2^8$ | 47.52 | 47.75 | -0.23 | 0.48 | 52.04 | 52.09 | -0.05 | 0.09 | 53.77 | 54.39 | -0.63 | 1.17 |
| $2^9$ | 50.04 | 50.44 | -0.40 | 0.80 | 53.84 | 54.80 | -0.96 | 1.79 | 56.67 | 58.13 | -1.46 | 2.58 |
| $2^{10}$ | 51.64 | 53.07 | -1.43 | 2.76 | 56.08 | 57.44 | -1.36 | 2.42 | 59.17 | 61.67 | -2.50 | 4.23 |

this section, we begin by analyzing the utilization efficiency of thinking tokens in Qwen3 models, followed by a comparison of two types of sequential scaling: multi-turn search and multi-turn reflection. Additional case studies are provided in Table 13. For implementation of token comparison, we don't truncate when the decoding response reaches a predefined threshold to restrict the response length of LLMs since it may harm the ability of LLMs. Instead, we generate $K$ responses and sum up the tokens used, and compared them under the same token budget.

**Inefficient Utilization of Thinking Tokens.** Qwen3 models support both "thinking" and "no thinking" modes (Yang et al., 2025b), allowing manual adjustment of the number of thinking tokens before the model produces a final answer. To investigate the influence of increasing thinking tokens in Self-Search settings, we conduct a comparative study evaluating the impact of whether enabling the thinking process during inference. We only count the token used out of `<search>`...`</search>`, `<information>`...`</information>`, and `<answer>`...`</answer>` for comparison of thinking token. As presented in Figure 7, the results demonstrate that as the number of assigned tokens increases, long CoT reasoning doesn't yield a better performance, contradictory to what is observed in complex math questions. This is probably attributed to that the solution to agentic search mainly relies on the usage of knowledge, either internal or external, rather than solely thinking. These findings indicate that short-CoT should be preferred in Self-Search settings to maximize token efficiency. We show the case study in Table 12.

**Multi-Turn Self-Search is inefficiency.** Following the established approach in search agent literature (Jin et al., 2025a; Sun et al., 2025) that formalizes search as a multi-turn process, we perform Self-Search for each rollout. Upon generating a search query, we prompt the model to provide relevant information for that query, incorporate this information into the reasoning context, and continue the iterative reasoning process until reaching a final decision. We denote the number of such interactions as $N$. The instruction for LLMs to provide relevant information is listed in Appendix A.1.1. Since our approach eliminates the need for external search engines (Google, Bing, etc.), we

Table 10: Comparison of actual vs. fitted values, residuals, and relative errors for Qwen2.5-14B-Instruct, Qwen2.5-72B-Instruct, and Qwen3-32B.

| | Qwen2.5-14B-Instruct | | | | Qwen2.5-72B-Instruct | | | | Qwen3-32B | | | |
|---|---|---|---|---|---|---|---|---|---|---|---|---|
| K | Actual | Fitted | Residual | Rel. Error (%) | Actual | Fitted | Residual | Rel. Error (%) | Actual | Fitted | Residual | Rel. Error (%) |
| $2^0$ | 28.10 | 30.91 | -2.81 | 10.01 | 24.40 | 28.39 | -3.99 | 16.34 | 26.96 | 29.16 | -2.20 | 8.17 |
| $2^1$ | 34.97 | 35.05 | -0.09 | 0.25 | 31.50 | 32.09 | -0.59 | 1.88 | 31.68 | 32.44 | -0.76 | 2.39 |
| $2^2$ | 39.83 | 39.22 | 0.62 | 1.55 | 36.10 | 35.85 | 0.25 | 0.69 | 36.64 | 35.75 | 0.89 | 2.43 |
| $2^3$ | 44.57 | 43.35 | 1.22 | 2.73 | 41.47 | 39.62 | 1.85 | 4.45 | 40.24 | 39.07 | 1.17 | 2.91 |
| $2^4$ | 48.87 | 47.41 | 1.46 | 2.98 | 45.83 | 43.36 | 2.47 | 5.39 | 43.44 | 42.37 | 1.07 | 2.45 |
| $2^5$ | 52.37 | 51.35 | 1.01 | 1.94 | 49.03 | 47.04 | 1.99 | 4.06 | 46.76 | 45.64 | 1.12 | 2.41 |
| $2^6$ | 55.83 | 55.15 | 0.68 | 1.22 | 51.80 | 50.63 | 1.17 | 2.26 | 49.64 | 48.83 | 0.81 | 1.62 |
| $2^7$ | 59.17 | 58.78 | 0.39 | 0.65 | 55.10 | 54.10 | 1.00 | 1.81 | 51.76 | 51.95 | -0.19 | 0.37 |
| $2^8$ | 61.80 | 62.22 | -0.42 | 0.68 | 57.00 | 57.44 | -0.44 | 0.77 | 54.48 | 54.97 | -0.49 | 0.91 |
| $2^9$ | 64.60 | 65.46 | -0.86 | 1.34 | 58.97 | 60.63 | -1.66 | 2.82 | 57.36 | 57.89 | -0.53 | 0.92 |
| $2^{10}$ | 66.73 | 68.50 | -1.77 | 2.65 | 60.80 | 63.66 | -2.86 | 4.70 | 59.48 | 60.69 | -1.21 | 2.03 |

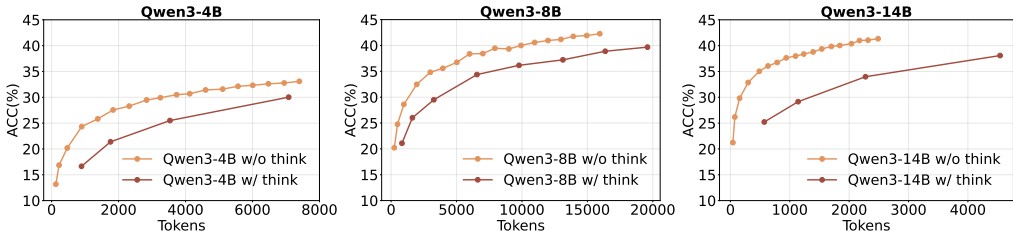

Figure 7: The performance of various sizes of Qwen3 averaged on six benchmarks, when enabled forced-thinking or not. The x-axis is measured by the number of tokens used by thinking solely.

avoid API costs and inference budget constraints typically associated with online search. We show the statistics of the search numbers required by each model across benchmarks in Table 11. Based on the table, we set $N = 10$ to ensure sufficient iterations for every sample to converge to a final answer. As shown in Figure 8, when measured by tokens consumed, naive repeated sampling shows better performance and a steady performance growth, further highlighting the upper bound of LLMs themselves as an implicit simulator of world knowledge.

Table 11: Statistics of self-search turns for different models across benchmarks.

| Model | Average Search Num | Max Search Num |
|---|---|---|
| Llama-3.2-3B-Instruct | 1.8 | 5.0 |
| Llama-3.1-8B-Instruct | 1.3 | 7.0 |
| Qwen2.5-3B-Instruct | 2.1 | 5.0 |
| Qwen2.5-7B-Instruct | 2.3 | 8.0 |

**Self-Search with Reflection is inefficiency.** The "Aha Moment", introduced by Deepseek-R1 (DeepSeek-AI, 2025), demonstrates emergent reflection and exploration capabilities in LLMs, particularly in math and code generation. To investigate whether this reflective behavior extends to information search tasks without external sources, we incorporate reflection-triggering phrases into our sampling process. Specifically, we append *"Wait, wait, wait"* after each generated response to encourage the model to reconsider and explore alternative reasoning paths. Figure 8 presents the experimental results. We also find that under the same token budget, reflection doesn't yield a better performance measured by `pass@k` compared with naive repeated sampling.

In conclusion, we find that increasing the number of thinking tokens and incorporating multi-turn generation are not always beneficial in Self-Search settings. This suggests that knowledge utilization may be more advantageous than reasoning in these scenarios. Further investigation is warranted, particularly in the context of language models as world models (Hao et al., 2023; Gu et al., 2024).

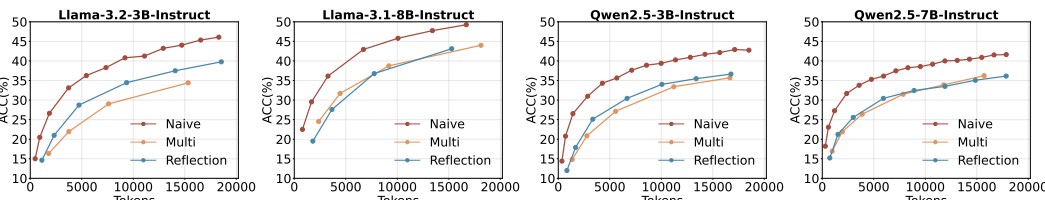

Figure 8: The performance of Repeated Sampling of Self-Search, Multi-turn Self-Search, and Self-Search with Reflection measured under the same token budget across four models.

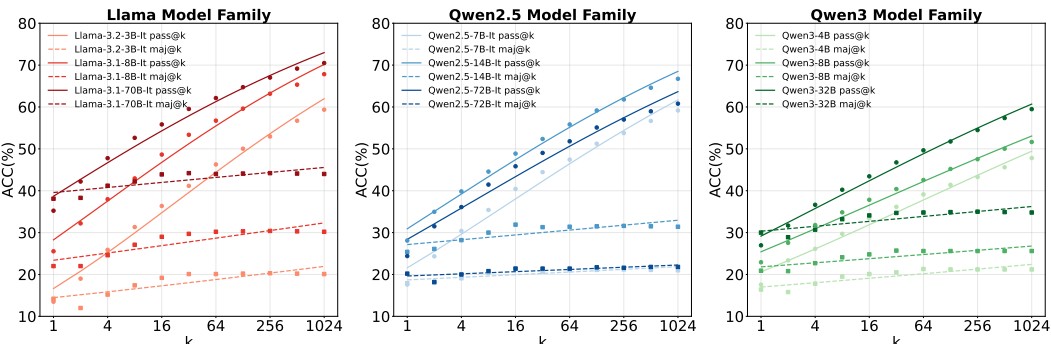

Figure 9: Majority voting results of different models averaged on six benchmarks.

## A.4 CASE STUDIES

### A.4.1 CASE STUDY FOR QWEN3 WITH/WITHOUT THINKING

We show a case study for Qwen3 with/without thinking mode in Table 12. When enabling thinking, the model will spend many tokens for reasoning at the beginning.

### A.4.2 CASE STUDY FOR MULTI-TURN AND REFLECTION REPEATED SAMPLING

We show the case study for multi-turn and reflection-based repeated sampling in Table 13.

## B SELF-SEARCH REINFORCEMENT LEARNING

### B.1 PROMPTS

We list the instruction used for direct reasoning in Table 14, the instruction for RAG in table 15, and the instruction for R1-like model training in Table 16.

### B.2 IMPLEMENTATION DETAILS

#### B.2.1 BASELINE IMPLEMENTATION

**Direct Answer and CoT** We show the instruction used in Appendix B.1 and 3.2 respectively. We set the temperature to 0.0 for consistent evaluation.

**RAG** We use the instruction listed in Appendix B.1. We use Jina and Google search. We set the temperature to 0.0 for consistent evaluation.

**R1** We use the instruction listed in Appendix B.1. We train for up to 5 epochs, and we stop training if any collapse is observed, including response length and training accuracy. We select the checkpoint with the best performance. The training batch size is 256, and the learning rate is 1e-6.

Table 12: Case study for Qwen3-32B with thinking and without thinking.

**Question:** Who is the mother of the director of film Polish-Russian War (Film)?
**Golden Answer:** Małgorzata Braunek
**Extracted Answer:** Elżbieta Munk

**Thinking Process (With Thinking):**
`<think>` Okay, let's see. The question is asking for the mother of the director of the film "Polish-Russian War." First, I need to figure out who directed that film. But wait, I don't remember any film specifically called "Polish-Russian War." Maybe it's known by a different title in English? Sometimes movies have different names in different countries.
Alternatively, maybe it's a historical film about the Polish-Russian conflicts. Let me think. There are several conflicts between Poland and Russia, like the Polish-Soviet War in the early 20th century. Could the film be about that? Maybe the user is referring to a film like "The Battle of Somosierra," but that's a different conflict. Hmm.
Wait, perhaps the user is referring to a Polish film. Maybe the director is someone well-known. Let me try to search for the film "Polish-Russian War" to find the director. If I can find the director, then I can look up their mother.
So, first step: search for "Polish-Russian War film director." Let's see what comes up. If there's a film by that name, I can get the director's name. If not, maybe there's confusion with the title. Alternatively, maybe it's a documentary or a less-known film.
...
So first, I need to search for "Polish-Russian War film director" to get the correct director's name. `</think>`
`<search>` Polish-Russian War film director `</search>`
`<information>` 1. "Bitwa Warszawska" (1975) directed by Andrzej Munk - focuses on the Polish-Soviet War. 2. "The First Echelon" (1968) directed by Georgy Yungvald-Khilkevich - Soviet film about Polish-Soviet War. 3. "The War" (2015) directed by Andrzej Munk - Polish film about the Polish-Soviet War. `</information>`
`<search>` Andrzej Munk mother `</search>`
`<information>` 1. Andrzej Munk's mother was Elżbieta Munk, a Jewish woman from Warsaw who perished in the Holocaust. `</information>`
`<search>` Georgy Yungvald-Khilkevich mother `</search>`
`<information>` 1. Limited information found; focus on his career in Soviet cinema. `</information>`
`<answer>` Elżbieta Munk `</answer>`

**Thinking Process (Without Thinking):**
To answer the question, I first need to identify the director of the film *Polish-Russian War* and then determine who their mother is. Let me start by finding out who directed the film.
`<search>` Who directed the film Polish-Russian War? `</search>`
`<information>` 1. The film "Polish-Russian War" (also known as "Polsko-Rosyjska Wojna") was directed by Andrzej Wajda. 2. Andrzej Wajda was a renowned Polish film director, screenwriter, and actor, known for his contributions to Polish cinema. `</information>`
Now that I know the director is Andrzej Wajda, I can proceed to find out who his mother was.
`<search>` Who was the mother of Andrzej Wajda? `</search>`
`<information>` 1. Andrzej Wajda's mother was Jadwiga Wajda (née Bujak). 2. Jadwiga Wajda was a Polish teacher and a significant influence in Andrzej Wajda's early life. `</information>`
`<answer>` Jadwiga Wajda `</answer>`

Table 13: Case study for Multi-turn and reflective Repeated Sampling.

**Question:** Were Scott Derrickson and Ed Wood of the same nationality?
**Golden Answer:** yes
**Extracted Answer:** No, Scott Derrickson and Ed Wood are not of the same nationality, with Derrickson being Canadian and Wood being American.

**Thinking Process (Multi):**
<think> I need to consider the nationality of Scott Derrickson and Ed Wood to determine if they are of the same nationality. Scott Derrickson is known for directing movies like "Doctor Strange" and "The Exorcism of Emily Rose". Ed Wood is known for directing cult classic films like "Plan 9 from Outer Space". To find their nationalities, I'll need to look into their birthplaces. </think>
<search> Scott Derrickson nationality </search>
<information> Doc 1: Scott Derrickson is a Canadian film director, screenwriter, and producer known for directing Marvel's Doctor Strange, released in 2016. Doc 2: Born in 1966, Derrickson grew up in Ontario, Canada, before pursuing a career in filmmaking. His background in animation and visual effects influenced his work. Doc 3: Derrickson's work often explores the intersection of faith and science. He co-created the TV series Eureka and wrote the screenplay for The Exorcism of Emily Rose. Doc 4: As a director, Derrickson's films often combine supernatural and psychological elements. His work has been praised for its unique visual style and atmospheric tension. Doc 5: Derrickson's production company, Film Science, has produced several films and TV series. He continues to explore the boundaries of storytelling through his work. </information>
<think> Now that I know Scott Derrickson is Canadian, I'll look into Ed Wood's nationality. </think>
<search> Ed Wood nationality </search>
<information> Doc 1: Ed Wood is an American filmmaker known for his low-budget horror and sci-fi films, including the cult classic Plan 9 from Outer Space. Doc 2: As a pioneer in the low-budget film industry, Wood's work often featured unique and bizarre storylines, captivating a niche audience. Doc 3: Born in Poughkeepsie, New York, Ed Wood began making films as a teenager and developed a passion for storytelling and filmmaking. Doc 4: Wood's most famous film, Plan 9 from Outer Space, is often cited as one of the worst films ever made, but also as a cult classic. Doc 5: Ed Wood's life and career have been the subject of numerous documentaries and books, exploring his legacy as a filmmaker and outsider artist. </information>
<answer> No, Scott Derrickson and Ed Wood are not of the same nationality, with Derrickson being Canadian and Wood being American. </answer>

**Thinking Process (Reflection):**
<think> I need to determine the nationality of Scott Derrickson and Ed Wood to compare them. </think>
<search> Scott Derrickson nationality </search>
<information> Scott Derrickson is an American film director, screenwriter, and cinematographer. He was born in the United States. </information>
<search> Ed Wood nationality </search>
<information> Ed Wood was an American film director, screenwriter, and actor. He was born in the United States. </information>
<answer> Yes, Scott Derrickson and Ed Wood are of the same nationality. They are both American. </answer>
*wait, wait, wait*
<search> nationality American </search>
<information> Americans are people born or citizens of the United States. </information>
<search> nationality same </search>
<information> The terms "same" and "nationality" typically refer to identical or similar ethnic or cultural identity. </information>
<answer> Yes, Scott Derrickson and Ed Wood are of the same nationality. They are both American. </answer>

Table 14: Instruction for Direct Reason.

Answer the given question. Provide the answer inside `<answer>` and `</answer>` without any additional information. For example, `<answer>` Beijing `</answer>`.

Table 15: Instruction for RAG.

You are a knowledgeable assistant that utilizes the provided documents to answer the user's question accurately.
Question: question
Documents: documents
Guidelines:
- Analyze the provided documents to extract relevant information. Synthesize the information to formulate a coherent and accurate answer.
- Ensure that your response directly addresses the user's question using the information from the documents.

We use a KL loss coef of 0.001. We train our models based on GRPO, and for each prompt, we generate 5 responses. For evaluation, we use temperature = 0.0.

**ZeroSearch**   We use the instruction listed in Appendix 3.2. We use the same setting as R1. The max turn is 2. The simulation LLM is `Simulation_LLM_google_14B`. The start threshold is 0.0, and the end threshold is 0.5.

**Search-R1**   We use the same setting as R1. When training, we use e5 as the retriever and Wikipedia as the corpus. When testing, we use Google Search for consistent comparison.

### B.2.2   OTHER ALGORITHM IMPLEMENTATION

**PPO**   We train for up to 5 epochs, and we stop training if any collapse is observed, including response length and training accuracy. We select the checkpoint with the best performance. The training batch size is 256, and the learning rate is 1e-6. We use a KL loss coef of 0.001. The learning rate for the critic model is 1e-5. We use a standard GAE as our advantage estimator, with $\gamma = 1.0$ and $\lambda = 1.0$.

**REINFORCE++**   We train for up to 5 epochs, and we stop training if any collapse is observed, including response length and training accuracy. We select the checkpoint with the best performance. The training batch size is 256, and the learning rate is 1e-6. We use a KL loss coef of 0.001.

**DAPO**   We train for up to 5 epochs, and we stop training if any collapse is observed, including response length and training accuracy. We select the checkpoint with the best performance. The training batch size is 256, and the learning rate is 1e-6. We don't use a KL loss. The low clip ratio is 0.2, and the high clip ratio is 0.28. We filter groups based on accuracy.

**KL-Cov**   We train for up to 5 epochs, and we stop training if any collapse is observed, including response length and training accuracy. We select the checkpoint with the best performance. The training batch size is 256, and the learning rate is 1e-6. We don't use a KL loss. We use a k-percent of 0.2.

Table 16: Instruction for R1 Training.

Answer the given question. Provide the answer inside `<answer>` and `</answer>`. For example, `<answer>` Beijing `</answer>`. Let's search step by step. You can break the question into pieces and answer one by one.

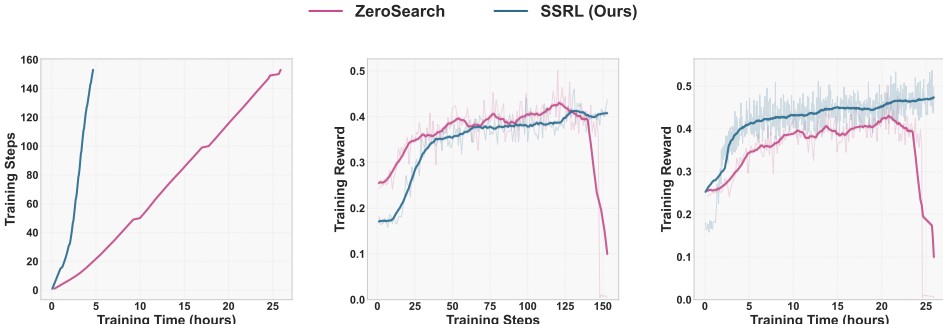

Figure 10: The training curve of Llama-3.1-8B-Instruct using SSRL and ZeroSearch. The first figure is the time used during training. The second figure is the training reward on the same training step, and the third figure is the training reward consuming the same time.

### B.2.3 TTRL

We set the max prompt length to 1024, and the max response length to 3076. The batch size is 8, and for each prompt, we rollout 32 times. The rollout temperature is set at 0.6. We use a learning rate of 5e-7 and a warm up step of 62. We remove the use of KL loss as in the original paper. We train for 80 epochs and stop when the performance converges.

### B.3 ABLATION STUDIES

### B.3.1 BENEFITS OF INFORMATION MASKING

Since all retrieved information originates from the reasoning model itself, jointly training the model on both the reasoning process and information generation represents a natural optimization strategy. To test the effectiveness of training full trajectories, we conduct experiments for training with and without information masking during the learning process. Figure 11 presents comparative results. The experimental results demonstrate that information masking consistently enhances model performance across benchmarks. Analysis of the training dynamics, which is listed in Appendix B.3.8, reveals that masking information tokens during training encourages the model to generate more comprehensive and detailed reasoning trajectories. The enhanced capability provides a compelling explanation for the consistent performance improvements observed across diverse question-answering tasks. By preventing the model from simply copying retrieved information during training, the masking strategy forces deeper engagement with the reasoning process itself, ultimately leading to more robust problem-solving abilities at inference time. Moreover, training without the information mask leads to a more serious entropy collapse which we attribute to the training of world knowledge token, hindering the exploration of RL process, further deteriorating the performance.

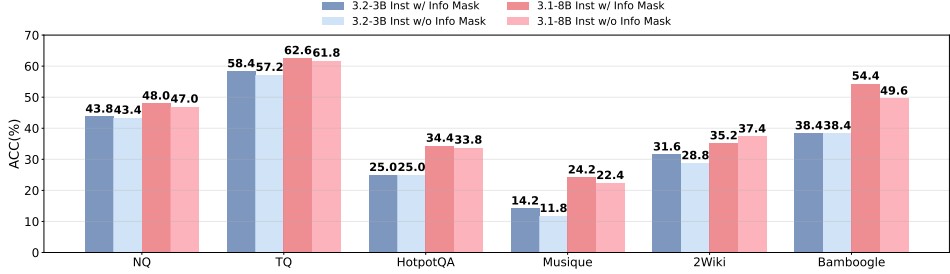

Figure 11: The performance of Llama-3.2-3B-Instruct and Llama-3.1-8B-Instruct when trained with and without the information mask.

### B.3.2 IMPACT OF FORMAT-BASED REWARD

To effectively elicit the dual capabilities of language models as both reasoners and internal search engines, we design a format reward that enforces adherence to our structured reasoning framework.

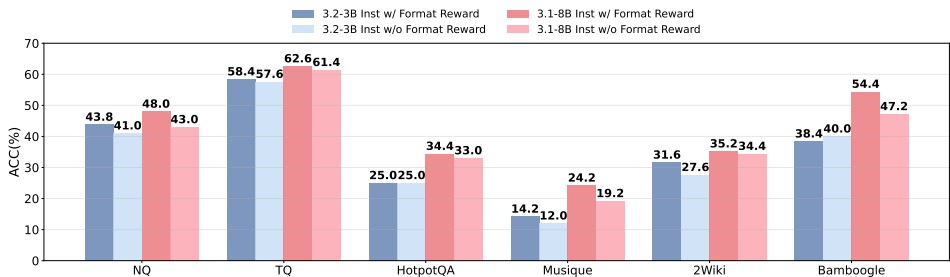

Figure 12: The performance of Llama-3.2-3B-Instruct and Llama-3.1-8B-Instruct when trained with and without format reward. All of the models are trained with an information mask.

Table 17: The performance of different $\lambda_f$.

| Temperature | GeneralQA | | Multi-HopQA | | | | Avg |
|---|---|---|---|---|---|---|---|
| | NQ | TQ | HotpotQA | Musique | 2Wiki | Bamboogle | |
| 0.1 | 43.8 | **58.4** | 25.0 | **14.2** | **38.4** | **31.6** | **35.2** |
| 0.2 | 41.8 | 57.2 | 26.8 | 12.2 | **40.0** | 26.6 | 34.1 |
| 0.4 | **42.2** | 58.0 | **26.8** | 12.8 | 36.8 | 26.4 | 33.9 |

This reward component ensures that models consistently follow the prescribed format of thinking, searching, and information gathering throughout their reasoning process. We evaluate the effectiveness of format reward through ablation studies comparing models trained with and without this component. Figure 12 presents the comparative results, demonstrating that format reward consistently improves performance for both base and instruction-tuned models across all benchmarks. These findings highlight that structured output formatting is crucial for successfully combining reasoning and search capabilities within a single model. The format reward acts as a critical scaffolding mechanism, guiding the model to maintain organized reasoning trajectories that facilitate effective internal knowledge retrieval. Without this structural guidance, models tend to produce less coherent reasoning paths that underutilize their internal search capabilities, resulting in degraded overall performance. Also, we observe a drastic response length growth without format reward, which indicates that format reward stabilizes the training process.

We further experiment on the effect of $\lambda_f$ on the final performance for comprehensive ablation. We show the experimental results in Table 17. We find that a larger $\lambda_f$ leads to inferior performance, which indicate that the outcome reward contributes most to the performance gain while the format reward is an auxiliary part which should distract the main optimization gradient..

### B.3.3 IMPORTANCE OF ON-POLICY SELF-SEARCH

In previous work, such as ZeroSearch (Sun et al., 2025), the fine-tuned LLM serves as an information provider. In contrast, we treat the policy model as an implicit simulator of world knowledge to supply information in above sections, which not only simplifies training but also significantly reduces training costs, particularly those associated with multi-turn rollouts.

To gain a comprehensive understanding, we examine two settings: one in which the information provider is the policy itself, and another in which the provider is the zero-step policy (i.e., a frozen policy). An information mask and a formatted reward are applied throughout all training procedures. We conduct experiments on four models from two different model families to evaluate their generalization capabilities. The results, presented in Table 19, reveal a dramatic collapse after approximately 100 training steps, with training rewards either remaining stagnant or decreasing sharply. We also observe significant performance degradation when using a frozen LLM as the information provider.

### B.3.4 EMPIRICAL STUDY ON HALLUCINATION

Here, we provide the analysis of the decreased hallucination through two perspectives. When training solely on the combination of NQ and HotpotQA, the performance on other out-of-distribution

benchmarks is also improved, which can be seen as a decrease of hallucination since QA tasks can be viewed as an indicator of hallucination (Bang et al., 2025; Guan et al., 2024). Besides, to provided a more direct way to observe the hallucination rate, we randomly sample 150 generated information through the self-search process and hire annotators to verify the factuality. Each information is annotated by two annotators and if any one of them find it counterfactual, it will be regarded as a negative. We present the results in Table 18 which shows the effectiveness of SSRL for reducing hallucination. Furthermore, we provide case studies in Table 34 to further validate our assumption.

Table 18: The factuality rate between base model and SSRL-trained model. The larger the factuality score, the better the model is.

| Model | Base Model Factuality | SSRL-Trained Factuality | Improvement |
|---|---|---|---|
| Qwen2.5-3B-Instruct | 22.70% | 44.00% | **+21.3 p.p.** |
| Llama-3.2-3B-Instruct | 39.30% | 64.70% | **+25.4 p.p.** |

### B.3.5 COMPATIBILITY WITH RL ALGORITHMS

We present the performances when training models with different algorithms, including PPO, GRPO, Reinforce++, DAPO, and KL-Conv. We use Llama-3.2-3B-Instruct and Llama-3.1-8B-Instruct as our backbones. The implementation details is listed in Appendix B.2.2. We present our results in Table 20. We observe a non-trivia performance gap between different training algorithms, with GRPO-based algorithms, e.g., GRPO, DAPO, etc, performs better than PPO and RE-INFORCE++. The superior performance of PPO is also observed in Sun et al. (2025), which proves the effectiveness of repeated rollouts for search agent training. It is worth noting that when trained with online engines like Google, the repeated rollouts will lead to greater cost. However, since we train models totally offline, more rolloouts may result in better performance without additional cost.

### B.3.6 MODEL FAMILY COMPARISON

Since Qwen is widely regarded as a stronger base model than Llama in math or code tasks, we aim to find out whether the conclusion holds when it relies on internal knowledge to answer the knowledge-intensive questions. We also use the default setting for training Qwen. All the training consists of the format, reward, and information token mask. The experimental results is listed in Table 21. Though we still observe the same training pattern as in Llama, for example, the scaling effect and the superior ability of instruct models, the absolute performance is relatively lower than Llama series, indicating that the ability of Qwen to serve as a simulator of world knowledge is not as good as Llama. The finding contradicts the trend in reasoning tasks, such as math and code generation, where Qwen is always thought of as the best base model to start.

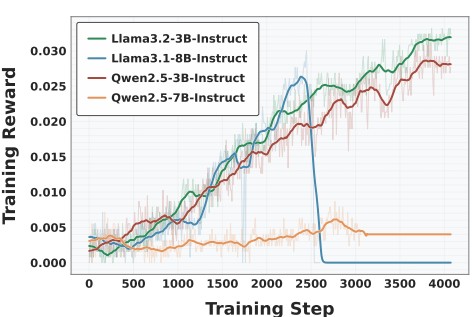

Figure 13: The performance of TTRL of various models on BrowseComp.

### B.3.7 COMPARISON BETWEEN GENERAL MODELS AND REASONING MODELS

LRMs show expressive performance on reasoning tasks like math and code generation. However, few work continues to train LRMs to adapt to other fields. To have a thorough overview, we compare the RL performance between general models and reasoning models. We use Qwen2.5 and Qwen3 for a comparison. The experimental results is shown in Table 22. We find that the performance of Qwen3 is generally lower than Qwen2.5. Recall in Figure 2, the upper bound of Qwen3 is also lower than Qwen2.5. These findings indicate that reasoning models trained with too much math or code generation data may be hard to transfer to other domains easily. We also notice an inferior instruction-following ability during our training process, resulting in a decreasing search number, which drops to 0 at a later training stage. However, this may also be attributed to the format reward of a certain prompt, which contradicts the initial tool call format of the Qwen3 series.

Table 19: Performance of Llama and Qwen2.5 with on-policy GRPO compared to freezing policy.

| Algorithm | GeneralQA | | Multi-HopQA | | | | Avg |
|---|---|---|---|---|---|---|---|
| | NQ | TQ | HotpotQA | Musique | 2Wiki | Bamboogle | |
| LLaMA-3.2-3B-Instruct | | | | | | | |
| GRPO | 43.8 | 58.4 | 25.0 | 14.2 | 31.6 | 38.4 | 35.2 |
| GRPO (Freezing) | 28.6 | 46.6 | 15.8 | 5.6 | 13.8 | 20.8 | 21.9 |
| Δ | −15.2 | −11.8 | −9.2 | −8.6 | −17.8 | −17.6 | −13.3 |
| | ↓34.7% | ↓20.2% | ↓36.8% | ↓60.6% | ↓56.3% | ↓45.8% | ↓37.8% |
| LLaMA-3.1-8B-Instruct | | | | | | | |
| GRPO | 48.0 | 62.6 | 34.4 | 24.2 | 35.2 | **54.4** | 43.1 |
| GRPO (Freezing) | 24.4 | 46.6 | 15.4 | 7.6 | 17.2 | 23.2 | 22.4 |
| Δ | −23.6 | −16.0 | −19.0 | −16.6 | −18.0 | −31.2 | −20.7 |
| | ↓49.2% | ↓25.6% | ↓55.2% | ↓68.6% | ↓51.1% | ↓57.4% | ↓48.0% |
| Qwen-2.5-3B-Instruct | | | | | | | |
| GRPO | 23.6 | 41.0 | 22.4 | 10.4 | 26.0 | 32.8 | 26.0 |
| GRPO (Freezing) | 9.8 | 18.4 | 7.4 | 5.0 | 7.5 | 12.8 | 10.2 |
| Δ | −13.8 | −22.6 | −15.0 | −5.4 | −18.5 | −20.0 | −15.8 |
| | ↓58.5% | ↓55.1% | ↓67.0% | ↓51.9% | ↓71.2% | ↓61.0% | ↓60.8% |
| Qwen-2.5-7B-Instruct | | | | | | | |
| GRPO | 31.4 | 44.4 | 26.0 | 11.8 | 31.0 | 36.8 | 30.2 |
| GRPO (Freezing) | 15.6 | 37.4 | 15.0 | 7.2 | 15.2 | 23.2 | 22.6 |
| Δ | −15.8 | −7.0 | −11.0 | −4.6 | −15.8 | −13.6 | −7.6 |
| | ↓50.3% | ↓15.8% | ↓42.3% | ↓39.0% | ↓51.0% | ↓37.0% | ↓25.2% |

| Algorithm | GeneralQA | | Multi-HopQA | | | | Avg |
|---|---|---|---|---|---|---|---|
| | NQ | TQ | HotpotQA | Musique | 2Wiki | Bamboogle | |
| LLaMA-3.2-3B-Instruct | | | | | | | |
| GRPO | 43.8 | 58.4 | 25.0 | **14.2** | 31.6 | **38.4** | **35.2** |
| DAPO | **44.6** | 58.0 | **26.8** | 12.8 | 26.6 | **38.4** | 34.5 |
| KL-Cov | 41.8 | **58.6** | 24.6 | 12.4 | 28.6 | **38.4** | 34.1 |
| REINFORCE++ | 42.2 | 55.8 | 25.6 | 12.6 | **32.0** | 30.8 | 33.2 |
| PPO | 35.0 | 55.8 | 21.8 | 11.4 | 29.6 | 30.4 | 30.7 |
| LLaMA-3.1-8B-Instruct | | | | | | | |
| GRPO | 48.0 | 62.6 | **34.4** | **24.2** | 35.2 | **54.4** | 43.1 |
| DAPO | **48.6** | 63.8 | **34.4** | 21.6 | 39.2 | 52.0 | **43.3** |
| KL-Cov | 44.8 | 63.6 | 32.6 | 22.8 | 37.4 | 52.8 | 42.3 |
| REINFORCE++ | 46.2 | **64.4** | 33.4 | 18.4 | **43.2** | 36.4 | 40.3 |
| PPO | 37.4 | 58.4 | 27.0 | 17.0 | 38.4 | 37.4 | 36.2 |

Table 20: The performance of Llama-3.2-3B-Instruct and Llama-3.1-8B-Instruct when trained with different RL algorithms. All of the models are trained with an information mask and a format reward.

### B.3.8 DYNAMICS OF TRAINING WITH AND WITHOUT INFORMATION MASK

We show the training dynamics with and without the information mask in Figure 14. The experimental results demonstrate that the information mask significantly enhances the model's search behavior activity. This indicates that the information mask mechanism encourages the model to perform more search operations, potentially improving the model's reasoning capabilities in complex tasks.

### B.3.9 GROUP SIZE ABLATION

For GRPO, we set the group size to 5 as in Jin et al. (2025b). In this part, we ablate on the impact of group size on the training dynamics and final performance. We train for 5 epochs and stop when the final performance converges, and select the checkpoints with the largest validation score. We use the default setting as mentioned above. We experiment on Qwen2.5-3B-Instruct and Llama-3.2-3B-Instruct. We show the training curve in Figure 14 and the results in Table 24. We observe a comparable performance when trained with a larger group size, but a faster convergence rate.

Table 21: The performance of Qwen2.5 models on General QA and Multi-Hop QA tasks.

| Model | GeneralQA | | Multi-HopQA | | | | Avg |
|---|---|---|---|---|---|---|---|
| | NQ | TQ | HotpotQA | Musique | 2Wiki | Bamboogle | |
| **Qwen2.5-3B-Instruct** | | | | | | | |
| Search-R1-base | 40.6 | 60.0 | 29.2 | 11.2 | 32.0 | 12.5 | 30.9 |
| Search-R1-inst | 35.8 | 55.8 | 33.2 | 7.6 | 26.0 | 12.5 | 28.5 |
| ZeroSearch-base | 43.0 | 61.6 | 33.8 | 13.0 | 34.6 | 13.9 | 33.3 |
| ZeroSearch-inst | 41.4 | 57.4 | 27.4 | 30.0 | 9.8 | 11.1 | 29.5 |
| SELF-SEARCH-BASE | 26.2 | 38.0 | 21.8 | 8.4 | 30.2 | 24.0 | 24.8 |
| SELF-SEARCH-INSTRUCT | 23.6 | 41.0 | 22.4 | 10.4 | 26.0 | 32.8 | 26.0 |
| **Qwen2.5-7B-Instruct** | | | | | | | |
| Search-R1-Base | 43.4 | 61.4 | 31.2 | 18.2 | 35.2 | 27.8 | 36.2 |
| Search-R1-Instruct | 42.4 | 63.4 | 32.8 | 17.4 | 33.2 | 26.4 | 35.9 |
| ZeroSearch-Base | 42.4 | 66.4 | 32.0 | 34.0 | 18.0 | 33.3 | 37.7 |
| ZeroSearch-Instruct | 43.6 | 65.2 | 34.6 | 18.4 | 35.2 | 27.8 | 37.5 |
| SELF-SEARCH-BASE | 28.8 | 44.2 | 25.0 | 11.4 | 30.4 | 35.2 | 29.0 |
| SELF-SEARCH-INSTRUCT | 31.4 | 44.4 | 26.0 | 11.8 | 31.0 | 36.8 | 30.2 |

Table 22: The performance of Qwen2.5 and Qwen3 models on General QA and Multi-Hop QA tasks.

| Model | GeneralQA | | Multi-HopQA | | | | Avg |
|---|---|---|---|---|---|---|---|
| | NQ | TQ | HotpotQA | Musique | 2Wiki | Bamboogle | |
| **Qwen2.5** | | | | | | | |
| Qwen2.5-3B-Instruct | 23.6 | 41.0 | 22.4 | 10.4 | 26.0 | 32.8 | 26.0 |
| Qwen2.5-7B-Instruct | 31.4 | 44.4 | 26.0 | 11.8 | 31.0 | 36.8 | 30.2 |
| **Qwen3** | | | | | | | |
| Qwen3-4B | 22.0 | 37.4 | 21.8 | 7.6 | 24.2 | 34.4 | 24.7 |
| Qwen3-8B | 27.0 | 45.2 | 27.0 | 10.8 | 31.8 | 36.0 | 29.6 |

### B.3.10 TEMPERATURE ABLATION

Considering the diversity of rollout matters a lot during RL process, we ablate on the choice the temperature. We show the experimental results on Llama-3.2-3B-Instruct in Table 23. From the table, we can observe that setting temperature at 1.0 achieves the best performance, while smaller temperature degrades the performance and larger temperature leads to training instability and collapse very soon.

### B.3.11 ADDITIONAL BENCHMARKS

For a comprehensive evaluation of self-contained search, we further evaluate on SimpleQA with Offline Search, Online Search (We drop out $K = 3$ for simplification), and Entropy-guided Search. We sample 200 records from SimpleQA. The results are listed in Table 26. We find that leveraging internal knowledge solely doesn't help complete tasks like SimpleQA (Wei et al., 2024), perhaps

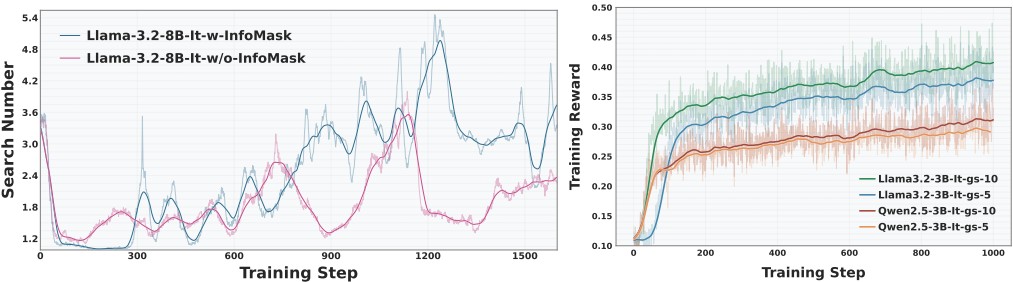

Figure 14: Left: Comparison of the search number with and without information mask on Llama-3.1-8B-Instruct. Right: Group size comparison.

Table 23: The performance of different temperature. The results of temperature greater than 1.0 is discarded since it collapses very soon.

| Temperature | GeneralQA | | Multi-HopQA | | | | Avg |
|---|---|---|---|---|---|---|---|
| | NQ | TQ | HotpotQA | Musique | 2Wiki | Bamboogle | |
| 1.0 | 43.8 | **58.4** | 25.0 | **14.2** | **38.4** | **31.6** | **35.2** |
| 0.6 | 41.8 | 58.0 | 25.8 | 13.2 | **38.4** | 28.0 | 34.2 |
| 0.4 | **44.0** | 56.4 | **28.0** | 12.6 | 36.8 | 28.4 | 34.3 |

Table 24: The performance of LLaMA and Qwen2.5 models trained with different group sizes.

| Model | GeneralQA | | Multi-HopQA | | | | Avg |
|---|---|---|---|---|---|---|---|
| | NQ | TQ | HotpotQA | Musique | 2Wiki | Bamboogle | |
| LLaMA-3.2-3B-Instruct | | | | | | | |
| Group Size = 5 | 43.8 | 58.4 | 25.0 | 14.2 | 31.6 | 38.4 | 35.2 |
| Group Size = 10 | 44.0 | 57.8 | 27.0 | 12.0 | 31.4 | 40.8 | 35.5 |
| Qwen2.5-3B-Instruct | | | | | | | |
| Group Size = 5 | 23.6 | 41.0 | 22.4 | 10.4 | 26.0 | 32.8 | 26.0 |
| Group Size = 10 | 26.2 | 37.8 | 22.6 | 8.4 | 27.0 | 24.8 | 24.5 |

due to SimpleQA is too challenging for models to retrieve factual knowledge from their parameters. However, when accessing the external knowledge base, our models still show great potential for such a complex task, indicating that SELF-SEARCH excels at organizing search queries and reasoning based on gathered information in real scenarios, even if trained totally in a simulated environment.

### B.3.12 ADDITIONAL RESULTS FOR SIM2REAL SEARCH

To test the importance of the first search, we experiment on two-stage generation, where we modify the generated response with the retrieved information to replace the first or the last information part, and then re-generate to obtain a final answer. The experiment results are shown in Table 25. It clearly demonstrates the importance of ensuring the quality of the first search and the corresponding information. That is, the first piece of search and the relevant information serves as an anchor for a successful search-and-answer trajectory. We further show the experimental results of entropy-guided search and Sim2Real search in Table 27.

### B.3.13 COMBINING SIMULATED SEARCH WITH REAL SEARCH

Our findings demonstrate that LLMs possess substantial internal knowledge, suggesting they should search externally only when necessary. Based on this insight, we propose an entropy-guided search strategy. For each generated sequence, we analyze the entropy trend of the initial search query: increasing entropy indicates model uncertainty, triggering external search; otherwise, we rely on internal knowledge. We use Sim2Real (All) as our baseline for fair comparison and always use external search for the first query based on the performance gains shown in Table 3 (see Appendix B.3.12 for ablation studies on first-search importance). We present our results in Table 27. The entropy-guided approach reduces search frequency by 20-42%, yielding substantial cost savings while maintaining performance comparable to full external search. As we observe above, though Llama-3.1-8B-Instruct fails under Sim2Real ($K = 3$), it achives better performance than Sim2Real (All), indicating that Llama-3.1-8B-Instruct is hard to leverage external information easily, which may be attributed to the gap between self-search gathered information and external one. These results reinforce our key finding: LLMs can effectively leverage their internal knowledge when they possess relevant information and know how to access it, making external search unnecessary in many cases.

### B.4 PERFORMANCE OF TEST-TIME RL

Considering unsupervised RL algorithms, e.g., TTRL, (Zuo et al., 2025), show great potential in math and code generation, we are curious about its generalization to SELF-SEARCH. We conduct

Table 25: The performance of LLaMA and Qwen2.5 models when replacing retrieved information at either the first or last search step using a real search engine.

| Model | GeneralQA | | Multi-HopQA | | | | Avg |
|---|---|---|---|---|---|---|---|
| | NQ | TQ | HotpotQA | Musique | 2Wiki | Bamboogle | |
| **LLaMA-3.2-3B-Instruct** | | | | | | | |
| Replace First | 44.4 | 63.4 | 34.8 | 17.2 | 37.8 | 42.4 | 40.0 |
| Replace Last | 41.0 | 59.6 | 24.8 | 12.8 | 32.2 | 39.2 | 34.9 |
| **LLaMA-3.1-8B-Instruct** | | | | | | | |
| Replace First | 39.4 | 55.8 | 34.0 | 26.8 | 39.8 | 53.6 | 41.6 |
| Replace Last | 47.4 | 62.2 | 34.4 | 22.2 | 39.0 | 49.6 | 42.5 |
| **Qwen2.5-3B-Instruct** | | | | | | | |
| Replace First | 33.8 | 49.6 | 28.2 | 12.0 | 33.6 | 28.0 | 30.9 |
| Replace Last | 23.2 | 37.0 | 22.8 | 7.4 | 29.2 | 35.5 | 25.9 |
| **Qwen2.5-7B-Instruct** | | | | | | | |
| Replace First | 35.8 | 56.6 | 34.0 | 17.0 | 34.8 | 40.8 | 36.5 |
| Replace Last | 28.2 | 45.2 | 25.4 | 11.4 | 30.2 | 28.8 | 28.2 |

Table 26: The performance of LLaMA and Qwen2.5 models on the SimpleQA task.

| Model | SimpleQA |
|---|---|
| **LLaMA-3.2-3B-Instruct** | |
| SSRL | 4.5 |
| Sim2Real | 45.5 |
| Entropy-guided Search | 43.0 |
| **LLaMA-3.1-8B-Instruct** | |
| SSRL | 7.0 |
| Sim2Real | 35.0 |
| Entropy-guided Search | 30.0 |
| **Qwen2.5-3B-Instruct** | |
| SSRL | 2.7 |
| Sim2Real | 51.0 |
| Entropy-guided Search | 45.0 |
| **Qwen2.5-7B-Instruct** | |
| SSRL | 4.0 |
| Sim2Real | 51.0 |
| Entropy-guided Search | 48.5 |

experiments on the Llama series, using the dataset consisting of NQ, TQ, HotpotQA, MusiQue, Bamboogle, 2WikiMultiHopQA, and BrowseComp [1]. The implementation details are listed in Appendix B.2.3. To ensure thorough analysis, we performed ablation studies both with and without information masking, while maintaining the format reward component, which remains essential for label voting mechanisms.

We measure the results using EM and show the experimental results in Table 29. We observe a better performance when trained with TTRL compared with GRPO. For Llama-3.2-3B-Instruct, the average performance is improved by 59%. This phenomenon indicates the generalization of TTRL across domains and model families. When using TTRL, we find that training without the information mask yields slightly better results, which contradicts RLVR. Surprisingly, we find that simply applying TTRL on the combined benchmarks results in a substantial improvement on BrowseC-

---

[1]For WebSailor, we sample 250 records from BrowseComp and evaluate them using a substring match. A response of WebSailor is considered right only if the generated prediction is in the ground truth or the ground truth is in the prediction.

Table 27: The performance of LLaMA and Qwen2.5 models when using either full or entropy-based selection over the real search engine. The average search number is the average number of `<search>` used during generation, i.e., online search plus self-search, if exists.

| Model | GeneralQA | | Multi-HopQA | | | | Avg | Avg. Search |
|---|---|---|---|---|---|---|---|---|
| | NQ | TQ | HotpotQA | Musique | 2Wiki | Bamboogle | | |
| LLaMA-3.2-3B-Instruct | | | | | | | | |
| Sim2Real (All) | 44.0 | 61.6 | 35.2 | 20.8 | 42.8 | 46.4 | 41.8 | 1.9 |
| Entropy-guided Search | 45.2 | 62.4 | 34.6 | 18.6 | 40.0 | 46.4 | 41.2 | 1.5 |
| LLaMA-3.1-8B-Instruct | | | | | | | | |
| Sim2Real-guided Search (All) | 39.6 | 54.6 | 34.6 | 25.0 | 36.8 | 50.4 | 40.2 | 2.6 |
| Entropy | 43.2 | 56.2 | 33.4 | 26.8 | 40.8 | 49.6 | 41.7 | 1.5 |
| Qwen2.5-3B-Instruct | | | | | | | | |
| Sim2Real (All) | 37.8 | 51.4 | 27.4 | 22.4 | 36.4 | 22.4 | 33.0 | 3.0 |
| Entropy-guided Search | 36.4 | 54.4 | 30.4 | 19.6 | 36.8 | 25.6 | 33.9 | 1.8 |
| Qwen2.5-7B-Instruct | | | | | | | | |
| Sim2Real (All) | 41.8 | 65.0 | 43.2 | 28.6 | 50.4 | 52.0 | 46.8 | 2.6 |
| Entropy-guided Search | 40.6 | 63.4 | 39.0 | 23.8 | 45.4 | 48.0 | 43.4 | 1.9 |

omp, even without external search engines. The accuracy curve on BrowseComp is presented in Figure 13, and the final performance metrics are summarized in Table 28.

There is an interesting observation that smaller models achieve higher scores on Browsecomp through TTRL, and when we delve into the cases, we find that these models prefer to point out an entity at first and check if it meets all the requirements, which is opposite to the search-and-answer paradigm. This further strengthens our opinion that LLMs contain information that once elicited, it can be applied to solve extremely complex questions.

Table 28: Performance on BrowseComp.

| Models | BrowseComp |
|---|---|
| WebSailor-3B | 2.0 |
| Qwen2.5-3B-Instruct (TTRL) | 3.9 |
| Qwen2.5-3B-Instruct (TTRL-Sim2Real) | 1.4 |
| Llama-3.2-3B-Instruct (TTRL) | **6.2** |
| Llama-3.2-3B-Instruct (TTRL-Sim2Real) | 4.1 |

We also experiment on Sim2Real on TTRL-trained models, and we show the results in Table 30. Though TTRL achieves better performance compared with RLVR, it introduces biases where models over-relying on its internal knowledge and are hard to adapt to real environments easily. We find that almost all queries are finished using one search query, even for BroweseComp. Therefore, in one-turn generation, the web search engine can't provide flexible information as the LLMs do. Moreover, we observe that TTRL-trained models prefer to select a candidate answer and verify it rather than search based on the question sequentially. We also find that it collapses frequently than RLVR, which is attributed to the unexpected deterministic behavior of policy models. We provide a case study in Table 35.

## B.5 CASE STUDIES

We provide case study of SSRL-trained Llama-3.2-3B-Instruct and SSRL-trained Qwen2.5-7B-Instruct in Table 31 and Table 32 for General QA and Multi-hop QA. After then, we show case study for Sim2Real Reasoning, providing the model with real search engine, in Table 33. We provide case study to demonstrate the decreased hallucination in Table 34. At last, we show case study for TTRL-trained models on BrowseComp in Table 35.

Table 29: The performance of Llama and Qwen trained with TTRL and GRPO. w/o info and w/ info indicate without information mask and with information mask, respectively. The largest value is denoted using **bold**.

| Algorithm | GeneralQA | | Multi-HopQA | | | | Avg |
|---|---|---|---|---|---|---|---|
| | NQ | TQ | HotpotQA | Musique | 2Wiki | Bamboogle | |
| **LLaMA-3.2-3B-Instruct** | | | | | | | |
| GRPO | 43.8 | 58.4 | 25.0 | 14.2 | 31.6 | 38.4 | 35.2 |
| TTRL (w/o info) | **58.6** | **76.4** | **47.2** | **37.2** | 59.4 | **57.6** | **56.1** |
| Δ | +14.8 | +18.0 | +22.2 | +23.0 | +27.8 | +19.2 | +20.9 |
| | ↑33.8% | ↑30.8% | ↑88.8% | ↑162.0% | ↑87.9% | ↑50.0% | ↑59.4% |
| TTRL (w/ info) | 57.4 | 74.0 | 45.2 | 36.4 | **60.2** | 56.0 | 54.9 |
| Δ | +13.6 | +15.6 | +20.2 | +22.2 | +28.6 | +17.6 | +19.7 |
| | ↑31.1% | ↑26.7% | ↑80.8% | ↑156.3% | ↑90.5% | ↑45.8% | ↑56.0% |
| **LLaMA-3.1-8B-Instruct** | | | | | | | |
| GRPO | 48.0 | 62.6 | 34.4 | 24.2 | 35.2 | **54.4** | 43.1 |
| TTRL (w/o info) | 43.0 | 64.0 | **35.6** | 27.2 | 47.0 | 52.0 | 44.8 |
| Δ | −5.0 | +1.4 | +1.2 | +3.0 | +11.8 | −2.4 | +1.7 |
| | ↓10.4% | ↑2.2% | ↑3.5% | ↑12.4% | ↑33.5% | ↓4.4% | ↑3.9% |
| TTRL (w/ info) | **49.2** | **67.4** | 35.4 | **40.2** | 48.2 | 52.0 | **48.7** |
| Δ | +1.2 | +4.8 | +1.0 | +16.0 | +13.0 | −2.4 | +5.6 |
| | ↑2.5% | ↑7.7% | ↑2.9% | ↑66.1% | ↑36.9% | ↓4.4% | ↑13.0% |
| **Qwen-2.5-3B-Instruct** | | | | | | | |
| GRPO | 23.6 | 41.0 | 22.4 | 10.4 | 26.0 | 32.8 | 26.0 |
| TTRL (w/o info) | **39.2** | **59.8** | **37.8** | **23.8** | **51.2** | **49.4** | **43.5** |
| Δ | +13.2 | +18.8 | +15.4 | +13.4 | +25.2 | +16.6 | +17.5 |
| | ↑55.9% | ↑45.9% | ↑68.8% | ↑128.8% | ↑96.9% | ↑50.6% | ↑67.3% |
| TTRL (w/ info) | 31.8 | 58.0 | 33.6 | 22.0 | 49.0 | 48.8 | 40.5 |
| Δ | +8.2 | +17.0 | +11.2 | +11.6 | +23.0 | +16.0 | +14.5 |
| | ↑34.7% | ↑41.5% | ↑50.0% | ↑111.5% | ↑88.5% | ↑48.8% | ↑55.8% |
| **Qwen-2.5-7B-Instruct** | | | | | | | |
| GRPO | 31.4 | 44.4 | 26.0 | 11.8 | 31.0 | 36.8 | 30.2 |
| TTRL (w/o info) | 40.6 | 63.2 | 40.4 | 28.8 | 53.2 | 64.0 | 48.4 |
| Δ | +9.2 | +18.8 | +14.4 | +17.0 | +22.2 | +27.2 | +18.2 |
| | ↑29.3% | ↑42.3% | ↑55.4% | ↑144.1% | ↑71.6% | ↑73.9% | ↑60.3% |
| TTRL (w/ info) | 34.6 | 54.8 | 32.6 | 20.2 | 43.0 | 50.4 | 39.3 |
| Δ | +3.2 | +10.4 | +6.6 | +8.4 | +12.0 | +13.6 | +9.1 |
| | ↑10.2% | ↑23.4% | ↑25.4% | ↑71.2% | ↑38.7% | ↑36.7% | ↑30.1% |

Table 30: Performance of Sim2Real Search Generalization on TTRL.

| Model | GeneralQA | | Multi-HopQA | | | | Avg |
|---|---|---|---|---|---|---|---|
| | NQ | TQ | HotpotQA | Musique | 2Wiki | Bamboogle | |
| **LLaMA-3.2-3B-Instruct** | | | | | | | |
| TTRL | **58.6** | **76.4** | **47.2** | **37.2** | **59.4** | **57.6** | **56.1** |
| Sim2Real | 56.6 | 74.8 | 46.0 | 36.0 | 59.0 | 54.4 | 54.5 |
| **Qwen2.5-3B-Instruct** | | | | | | | |
| TTRL | 39.2 | 59.8 | 37.8 | **23.8** | 51.2 | **49.4** | **43.5** |
| Sim2Real | **39.8** | **61.2** | **40.2** | 22.8 | **51.8** | 41.6 | 42.9 |

## C  FORMAT REWARD CODE

In this section, we provide the code to judge whether the rollout sequence is in a desired format for a search agent.

Listing 1: Format Reward Code.

```python
def format_reward(response: str) -> float:
```

Table 31: Case Study for SELF-SEARCH-Llama-3.2-3B-Instruct on General QA and Multi-hop QA.

**Question:** Who got the first nobel prize in physics?
**Answer:** Wilhelm Conrad Röntgen
**Thinking Process:**
`<think>` The question is asking who received the first Nobel Prize in Physics. I need to find out who that is. `</think>`
`<search>` first Nobel Prize in Physics `</search>`
`<information>` 1. The first Nobel Prize in Physics was awarded in 1901 to Wilhelm Conrad Röntgen for his discovery of X-rays. 2. The first Nobel Prize in Physics was awarded in 1901 to Wilhelm Conrad Röntgen. 3. The first Nobel Prize in Physics was awarded in 1901 to Wilhelm Conrad Röntgen for his discovery of X-rays. `</information>`
`<think>` The first Nobel Prize in Physics was awarded in 1901 to Wilhelm Conrad Röntgen for his discovery of X-rays. `</think>`
`<answer>` Wilhelm Conrad Röntgen `</answer>`

**Question:** If both Dictyosperma and Huernia are described as a genus
**Answer:** Yes
**Thinking Process:**
`<think>` The question is asking if both Dictyosperma and Huernia are described as a genus. `</think>`
`<search>` Dictyosperma genus `</search>`
`<information>` 1. Dictyosperma is a genus of plants in the asparagus family, Asparagaceae. 2. Dictyosperma is indeed a genus of plants. `</information>`
`<think>` Now we need to check if Huernia is also a genus. `</think>`
`<search>` Huernia genus `</search>`
`<information>` 1. Huernia is a genus of succulent plants in the family Apocynaceae. 2. Huernia is indeed a genus of plants. `</information>`
`<think>` Both Dictyosperma and Huernia are described as a genus. `</think>`
`<answer>` Yes `</answer>`

**Question:** Who plays the mouse in Alice in Wonderland
**Answer:** Barbara Windsor
**Thinking Process:**
`<think>` The question is asking who plays the mouse in Alice in Wonderland. I need to find out who plays the mouse in the 2010 film adaptation of Alice in Wonderland. `</think>`
`<search>` who plays the mouse in alice in wonderland `</search>`
`<information>` 1. According to Wikipedia, the mouse in the 2010 film Alice in Wonderland is played by Barbara Windsor. 2. According to IMDb, the mouse in the 2010 film Alice in Wonderland is played by Barbara Windsor. 3. According to Wikipedia, the mouse in the 2010 film Alice in Wonderland is played by Barbara Windsor. `</information>`
`<think>` The mouse in the 2010 film Alice in Wonderland is played by Barbara Windsor. However, I need to verify if this information is correct. `</think>`
`<search>` alice in wonderland mouse `</search>`
`<information>` 1. According to IMDb, the mouse in the 2010 film Alice in Wonderland is played by Barbara Windsor. 2. According to Wikipedia, the mouse in the 2010 film Alice in Wonderland is played by Barbara Windsor. 3. According to IMDb, the mouse in the 2010 film Alice in Wonderland is played by Barbara Windsor. `</information>`
`<think>` The mouse in the 2010 film Alice in Wonderland is played by Barbara Windsor. `</think>`
`<answer>` Barbara Windsor `</answer>`

Table 32: Case Study for SELF-SEARCH-Qwen2.5-7B-Instruct on General QA and Multi-hop QA.

**Question:** When was the last time Philadelphia won the Super Bowl?
**Golden Answers:** Super Bowl LII, 2017
**Extracted Answer:** 2017
**Solution Process:**
`<think>` First, we need to identify the last Super Bowl win for the Philadelphia team. `</think>`
`<search>` last time philadelphia won the superbowl `</search>`
`<information>`
- The Philadelphia Eagles last won the Super Bowl in 2017, defeating the New England Patriots in Super Bowl LII.
- The Philadelphia Eagles won Super Bowl LII 41-33.
- The last time the Philadelphia Eagles won the Super Bowl was in 2017.
`</information>`
`<think>` Based on the information, the last time the Philadelphia Eagles won the Super Bowl was in 2017. `</think>`
`<search>` year philadelphia eagles last won superbowl `</search>`
`<information>`
- The Philadelphia Eagles last won the Super Bowl in 2017.
- The 2017 Super Bowl was Super Bowl LII, which the Eagles won.
- The Philadelphia Eagles have not won a Super Bowl since 1980, but their last win was in 2017.
`</information>`
`<think>` Confirming the last Super Bowl win for the Philadelphia Eagles is indeed in 2017. `</think>`
`<search>` last superbowl win philadelphia team `</search>`
`<information>`
- The Philadelphia Eagles last won the Super Bowl in 2017.
- The last time the Philadelphia 76ers won an NBA championship was in 1983.
- The last time the Philadelphia Flyers won the Stanley Cup was in 1975.
`</information>`
`<think>` Based on all the information, the last time the Philadelphia Eagles won the Super Bowl was in 2017. `</think>`
`<answer>` 2017 `</answer>`

**Question:** Who is older, Aryeh Ben-Eliezer or Jason Pociask
**Answer:** Aryeh Ben-Eliezer
**Thinking Process:**
`<think>` First, we need to determine the birth years of both individuals to compare their ages. `</think>`
`<search>` Who is older, Aryeh Ben-Eliezer or Jason Pociask `</search>`
`<information>`
- Aryeh Ben-Eliezer was born on 15 October 1937.
- Jason Pociask was born on 19 July 1971.
`</information>`
`<think>` Now we know the birth years of both individuals. Aryeh Ben-Eliezer was born in 1937, while Jason Pociask was born in 1971. Clearly, Aryeh Ben-Eliezer is older. `</think>`
`<search>` Birth year of Aryeh Ben-Eliezer `</search>`
`<information>`
- Aryeh Ben-Eliezer was born on 15 October 1937. `</information>`
`<think>` Confirming the birth year of Aryeh Ben-Eliezer, we can now definitively state that he is older than Jason Pociask. `</think>`
`<answer>` Aryeh Ben-Eliezer `</answer>`

Table 33: Case Study for integration with a real search engine.

**Question:** Who was president of the United States in the year that Citibank was founded?
**Answer:** James Madison
**Ground Truth:** james madison
**Thinking Process:**
`<think>` First, we need to determine the year Citibank was founded to identify the president at that time. `</think>`
`<search>` Year Citibank was founded `</search>`
`<information>` - Citibank was founded in 1812 as City Bank of New York, and later became First National City Bank of New York. `</information>`
`<think>` Now that we know Citibank was founded in 1812, we need to find out who was the president of the United States in 1812. `</think>`
`<search>` Who was president in 1812 United States `</search>`
`<information>` - James Madison (March 16, 1751 [O.S. March 5, 1750] – June 28, 1836) was an American statesman, diplomat, and Founding Father who served as the fourth. `</information>`
`<think>` James Madison was the president in 1812, the year Citibank was founded. `</think>`
`<search>` James Madison presidency year 1812 `</search>`
`<information>` - Although he was accused of weakness in dealing with France and England, Madison won the presidency in 1808 by publishing his vigorous diplomatic dispatches. - American statesman, diplomat, and Founding Father who served as the fourth president of the United States from 1809 to 1817. - Fourth President 1809-1817. Fast Fact: James Madison, "Father of the Constitution," led the inconclusive War of 1812. `</information>`
`<think>` James Madison was the president from 1809 to 1817, and he was the president during the year Citibank was founded in 1812. `</think>`
`<search>` Who was president in 1812 United States `</search>`
`<information>` - James Madison was the fourth president of the United States, serving from 1809 to 1817. `</information>`
`<think>` Based on the information, James Madison was the president of the United States in 1812 when Citibank was founded. `</think>`
`<answer>` James Madison `</answer>`

**Question:** What physical force is used by the machine that extracts honey from honeycombs?
**Answer:** centrifugal force
**Thinking Process:**
`<think>` First, we need to identify the machine used for extracting honey and the physical force it uses. `</think>`
`<search>` machine used to extract honey from honeycombs physical force `</search>`
`<information>` - A honey extractor extracts the honey from the honey comb without destroying the comb. Extractors work by centrifugal force. A drum or container holds a frame. - A honey extractor is a device used to extract honey from the comb by spinning it at high speed. The centrifugal force causes the honey to be flung out of. - Honey extractors are simple mechanical devices that extract honey from honeycombs. They use centrifugal force in a drum or container with frame. `</information>`
`<think>` Based on the information, the machine used to extract honey from honeycombs uses centrifugal force. `</think>`
`<search>` physical force used in honey extractor `</search>`
`<information>` - A honey extractor is a piece of equipment used by beekeepers in the extraction of honey from honeycombs. A honey extractor works by using centrifugal force. - Extractors work by centrifugal force. A drum or container holds a frame basket which spins, flinging the honey out. With this method the wax comb stays intact. - Drum. The drum is the central component that holds the frames. It spins to create centrifugal force, which helps remove honey from the comb. `</information>`
`<think>` It is clear that the physical force used in a honey extractor is centrifugal force. `</think>`
`<search>` centrifugal force in honey extractor `</search>`
`<information>` - Extractors work by centrifugal force. A drum or container holds a frame basket which spins, flinging the honey out. With this method the wax comb stays intact. - A honey extractor uses centrifugal force to extract honey from the honey comb without destroying the comb. `</information>`
`<answer>` centrifugal force `</answer>`

Table 34: Case study for reduced hallucination. The bolded text is the factual information corresponding to the question.

---

**Question:** What body of water does the river that Toongabbie Creek flows into flow into?
**Base Model**: It flows into the Hawkesbury River.
**SSRL Model**: Toongabbie Creek flows into the Parramatta River. The Parramatta River flows into **Sydney Harbour**.

---

**Question:** Who wrote the TV Series containing the Finale episodes?
**Base Model**: The top result from a search engine is: "The Sopranos" finale was written by David Chase.
**SSRL Model**: The finale of Smallville was written by **Alfred Gough and Miles Millar**.

---

**Question:** Who is the mother of Katharine Smith Salisbury's brother?
**Base Model**: William Smith Salisbury was the son of William Henry Smith and Katharine Smith Salisbury.
**SSRL Model**: **Lucy Mack Smith** (July 8, 1775 – May 14, 1856) was the mother of Joseph Smith, founder of the Latter Day Saint movement. She is noted for writing the memoir,

---

```python
    response = response.strip()

    # Check if any tag content contains disallowed tags
    allowed_tags = {'think', 'search', 'information', 'answer', '/think',
        '/search', '/information', '/answer'}
    all_tags = re.findall(r'<([^>]+)>', response)
    for tag in all_tags:
        if tag not in allowed_tags:
            return 0.0

    # Must start with <think> and end with </answer>
    if not (response.startswith('<think>') and response.endswith('</
        answer>')):
        return 0.0

    # Extract all tags in order
    tags = re.findall(r'<(/?(?:think|search|information|answer))>',
        response)

    # Check if any tag content is empty
    tag_contents = {
        'think': re.findall(r'<think>(.*?)</think>', response, re.DOTALL)
            ,
        'search': re.findall(r'<search>(.*?)</search>', response, re.
            DOTALL),
        'information': re.findall(r'<information>(.*?)</information>',
            response, re.DOTALL),
        'answer': re.findall(r'<answer>(.*?)</answer>', response, re.
            DOTALL)
    }

    if len(tags) < 4:
        return 0.0
    # Return 0 if any tag has empty content
    for tag_type, contents in tag_contents.items():
        for content in contents:
            if not content.strip():
                return 0.0
            if tag_type == 'search' and len(content.split('\n')) != 1:
                return 0.0
            if tag_type == 'search' and 'your query' in content.lower():
                return 0.0
```

Table 35: Case study of TTRL-trained Llama-3.2-3B-Instruct on BrowseComp.

---

**Question:** A professional football player completed a transfer to a new club in the 2010s, in exchange for another player. This player later described his time at the new club negatively, and seven articles were written about him by Football Insider in the years 2016 and 2017. He was released from his contract by a club founded in the 1920s more than 3 years after this transfer. He entered a new coaching role in 2020 at a football club in a different country known for their attacking style. The player for which he was exchanged ended his career at a football club representing a town that has a castle. How many league goals did the first player score for the club that he left in this transfer?

**Golden Answer:** 76

**Extracted Answer:** 76

**Thinking Process:**

`<think>` I need to find the professional football player who completed the transfer in the 2010s. I will search for the player's name. `</think>`

`<search>` Professional football player transferred in 2010s `</search>`

`<information>` The player is Daniel Sturridge, who transferred from Manchester City to Liverpool in 2013. He was released by Liverpool in 2019. He entered a new coaching role in 2020 at Middlesbrough. He ended his career at Middlesbrough, a football club representing a town with a castle. According to his career statistics, Daniel Sturridge scored 76 league goals for Manchester City. `</information>`

`<think>` Since I found the player's name, I can now find the number of league goals he scored for the club that he left. `</think>`

`<answer>` 76 `</answer>`

---

**Question:** A professional football player completed a transfer to a new club in the 2010s, in exchange for another player. This player later described his time at the new club negatively, and seven articles were written about him by Football Insider in the years 2016 and 2017. He was released from his contract by a club founded in the 1920s more than 3 years after this transfer. He entered a new coaching role in 2020 at a football club in a different country known for their attacking style. The player for which he was exchanged ended his career at a football club representing a town that has a castle. How many league goals did the first player score for the club that he left in this transfer?

**Golden Answer:** 76

**Extracted Answer:** 105

**Thinking Process:**

`<think>` I need to find the professional football player who completed the transfer in the 2010s. I will search for the player's name. `</think>`

`<search>` Professional football player transferred in 2010s `</search>`

`<information>` Heading into the 2010s, Cristiano Ronaldo's €94 million move from Manchester United to Real Madrid was the world record transfer. This list of the top 10 best college football transfers dating back to 2010 to come into creation from best to worst, beginning from one to ten. `</information>`

`<think>` Since I found the player, I can now find the club he was released from and the coaching role he entered. I can also find the club he ended his career at. I will search for the number of league goals he scored for the club that he left. `</think>`

`<answer>` 105 `</answer>`

---

```
            if tag_type == 'think' and 'your thoughts' in content.lower()
                :
                return 0.0
            if tag_type == 'answer' and 'your answer' in content.lower():
                return 0.0
            if tag_type == 'information' and 'your information' in
                content.lower():
                return 0.0

    # Check structure
    if tags[0] != 'think' or tags[1] != '/think':
        return 0.0

    if tags[-2] != 'answer' or tags[-1] != '/answer':
        return 0.0

    # Check search-information pairing in the middle
    middle_tags = tags[2:-2]  # Exclude initial think and final answer

    i = 0
    while i < len(middle_tags):
        if middle_tags[i] == 'search':
            # Must be followed by /search, information, /information
            if (i + 3 >= len(middle_tags) or
                middle_tags[i + 1] != '/search' or
                middle_tags[i + 2] != 'information' or
                middle_tags[i + 3] != '/information'):
                return 0.0
            i += 4
        else:
            i += 1

    think_num = response.count('<think>')
    search_num = response.count('<search>')
    information_num = response.count('<information>')
    if search_num != information_num:
        return 0.0

    max_turn = 2
    score = 1.0 / max_turn * think_num
    ratio = 1.0

    upper_bound = 8
    if think_num != search_num + 1:
        ratio = min(think_num, search_num + 1) / max(think_num,
            search_num + 1)

    return min(score, 1.0) * ratio if think_num <= upper_bound else 0.0
```