# OpenReview forum: "SSRL: Self-Search Reinforcement Learning"
_ICLR.cc/2026/Conference — Submitted to ICLR 2026_

### Official Review · Reviewer_Xsoz · 2025-10-27

**Soundness:** 2
**Presentation:** 4
**Contribution:** 3
**Rating:** 4
**Confidence:** 4

**Summary:**

The paper proposes SSRL (Self-Search Reinforcement Learning), a method that trains language models to perform self-generated search before answering questions.
Instead of using a real search engine, the model creates its own “search” and “information” steps inside the output, and then receives a reward based on the answer quality and output format.
The goal is to teach the model to reason and retrieve information more systematically, while reducing the need for real web queries.
The authors evaluate SSRL on several QA benchmarks using models from 3B to 70B parameters, and also test a Sim-to-Real setting where the model interacts with real search results.

**Strengths:**

1) Novelty:
The paper combines reinforcement learning with self-constructed search reasoning.
This idea of training an LLM in a simulated search environment is novel and interesting.

2) Cost efficiency:
Because it does not depend on real API calls or human feedback, the method is low-cost and scalable for small and medium-sized models.

3) Sim-to-Real transfer:
The results show that a model trained only with self-generated searches can still work reasonably well when connected to a real search engine.
This indicates the learned policy is relatively stable and generalizable.

4) Practical relevance:
The work explores how smaller models can approach the performance of larger ones through structured reasoning and sampling, which is useful for efficient deployment.

**Weaknesses:**

1) Methodology clarity:
The paper does not clearly define the RL setup — it is unclear what the exact states, actions, and rewards are.
The training pipeline (sampling, reward, update) is not fully described, and the “self-search” mechanism is hard to reproduce.
Although the code is provided, it is difficult to directly examine every detail in code.

2) Reward design limitations:
The reward is simple and rule-based, combining a binary “outcome reward” (correct / incorrect) and a “format reward” for structural tags.
This design is very discrete and task-specific, which can make learning unstable and limit generalization.
It makes the knowledge of LLM is trapped in its own training environment, and implicitly put a requirement on the training dataset.

3) Claims without quantitative proof:
The paper often claims that SSRL reduces hallucination or improves reasoning quality, but there is no direct measurement (e.g., factual consistency, hallucination rate, or human study).
The evidence is only indirect from QA accuracy, which is not enough.

4) Lack of ablation and analysis:
There is no ablation to test the importance of each reward term or design choice.
It is unclear how much improvement comes from SSRL itself versus the sampling or instruction tuning.

5) Task-specific scope:
The method is only evaluated on QA datasets, and it is uncertain whether the same idea can generalize to open-ended reasoning or dialogue tasks.
It's necessary to quantify the scope.

**Questions:**

Can the authors give more details about how the RL process is implemented?
For example, how are advantages computed, how is KL controlled, and how many updates per batch?

How sensitive is SSRL to the reward coefficients (e.g., λf)?
Would smoother or continuous rewards improve stability?

Did the authors try any direct hallucination or factual consistency evaluation?
This would help verify the claimed benefits.

How much of the gain comes from the structured format reward versus the self-search policy itself?
An ablation would clarify this.

Could this method be applied to other tasks beyond QA, such as summarization or dialogue?
If so, what changes would be needed in the reward or structure?

---

> ### Author Response · Authors · 2025-11-17
>
> **W1: RL in LLM vs. Traditional RL**
>
> Thank you for your question. SSRL is RL in LLM, not traditional RL. And we will detail the definition of LLM RL, and Agentic RL below.
>
> **LLM RL:**
> - **State:** The prompt together with the tokens generated so far.
> - **Action:** Depending on the granularity, the action may be an entire sequence 𝑦 (sequence-level), a token $a_t$ ∈ $\mathcal{V}$ (token-level), or a segment $y^{(k)} = (y_1^{(k)},y_2^{(k)}...,y_{T_k}^{(k)})$ (step-level) where k is the total length of the step.
> - **Reward:** Assigned based on the action granularity, e.g., sequence-level $𝑅(𝑥, 𝑦)$ at trajectory end, token-level $R_t = R(x, a_{(1:t)})$ per token, or step-level $R_k = R(x,y_{(1:k)})$per segment.[1]
>
> **Agentic RL:**
> - **State:** The environment feedback, like the searched information.
> - **Action:** Defined as $\mathcal{A} _ {\text{agent}}$ = $\mathcal{A} _ {\text{text}} \cup \mathcal{A} _ {\text{action}}$, where $\mathcal{A} _ {\text{text}}$ is the space of free-form natural language tokens like CoT and $\mathcal{A} _ {\text{action}}$ is the space of abstract, non-linguistic actions to interact with the environment.
> - **Reward:** Based on the downstream task, the rewards allow dense, sparse, or learned rewards. [2]
>
> In our setting, we convert agentic RL with LLM RL, using a simulated environment. So the state is the prompt together with the tokens generated. The action is the entire sequence, and the reward is the sequence-level reward. If the format passes the regex test, it will receive a format reward. If the generated answer matches the ground truth, it will receive an outcome reward.
>
> **Training Pipeline (GRPO):**
> - **Algorithm:** We use Group Relative Policy Optimization (GRPO) [3].
> - **Update:** We use an on-policy setting where the global batch size (256) equals the mini-batch size (256), resulting in one update per batch. The KL-loss coefficient ($\beta$ in Equation 5) is set to 0.001 (Line 323).
> - **Advantage Calculation:** Advantage is calculated using the GRPO formula shown in Equation 5.
> - **Sampling:** Rollouts are accelerated using VLLM. A rollout involves the policy attempting to solve a given question from the training set (NQ/HotpotQA).
>
> Moreover, we will provide you with a bash file to reproduce our results.
>
> ```bash
> set -x
> train_files="['Your Train Dataset']"
> test_files="[Your Test Dataset']"
> python3 -m verl.trainer.main_ppo \
>     algorithm.adv_estimator=grpo \
>     data.train_files="$train_files" \
>     data.val_files="$test_files" \
>     data.train_batch_size=256 \
>     data.max_turns=3 \
>     data.max_start_length=2048 \
>     data.max_prompt_length=1024 \
>     data.max_infer_length=500 \
>     data.max_obs_length=500 \
>     data.max_response_length=4096 \
>     actor_rollout_ref.model.path=Your policy \
>     actor_rollout_ref.actor.optim.lr=1e-6 \
>     actor_rollout_ref.model.use_remove_padding=True \
>     actor_rollout_ref.actor.ppo_mini_batch_size=256 \
>     actor_rollout_ref.actor.ppo_micro_batch_size_per_gpu=16 \
>     actor_rollout_ref.actor.use_kl_loss=True \
>     actor_rollout_ref.actor.kl_loss_coef=0.001 \
>     actor_rollout_ref.actor.kl_loss_type=low_var_kl \
>     actor_rollout_ref.actor.entropy_coeff=0 \
>     actor_rollout_ref.rollout.enable_chunked_prefill=False \
>     actor_rollout_ref.model.enable_gradient_checkpointing=True \
>     actor_rollout_ref.actor.fsdp_config.param_offload=True \
>     actor_rollout_ref.actor.optim.lr_warmup_steps_ratio=0.02 \
>     actor_rollout_ref.actor.fsdp_config.optimizer_offload=True \
>     actor_rollout_ref.rollout.log_prob_micro_batch_size_per_gpu=16 \
>     actor_rollout_ref.rollout.tensor_model_parallel_size=1 \
>     actor_rollout_ref.actor.enable_information_loss_mask=True \
>     actor_rollout_ref.rollout.name=vllm \
>     actor_rollout_ref.rollout.gpu_memory_utilization=0.7 \
>     actor_rollout_ref.rollout.n=5 \
>     actor_rollout_ref.ref.log_prob_micro_batch_size_per_gpu=16 \
>     actor_rollout_ref.ref.fsdp_config.param_offload=True \
>     algorithm.use_kl_in_reward=False \
>     trainer.critic_warmup=0 \
>     trainer.logger=['console','wandb'] \
>     trainer.n_gpus_per_node=8 \
>     +trainer.val_only=false \
>     trainer.nnodes=1 \
>     trainer.save_freq=10000 \
>     trainer.test_freq=16 \
>     trainer.sim2real=false \
>     trainer.type="qwen-7b" \
>     trainer.use_entropy=false \
>     trainer.total_epochs=5 $@

---

> > ### Author Response · Authors · 2025-11-17
> >
> > **W2: Reward Design and Generalization**
> >
> > We respectfully disagree that this reward design introduces instability or limits generalization.
> >
> > - Our use of a discrete reward structure (outcome + format) is consistent with other LLM-RL and Agentic-RL works, including DeepSeek-R1[4], Search-R1[5], and the Qwen3[6] series. Our results confirm that this combination yields superior performance and stability.
> >
> > - The model's knowledge is not "trapped." The goal of SSRL is to refine the model's internal knowledge utilization. As demonstrated by the improved performance on the OOD benchmarks, the skills acquired through training on NQ and HotpotQA generalize robustly.
> >
> > - SSRL trains the model to act as a "World Model" (similar to Meta's Code World Model[7]), enabling it to effectively leverage its internal parametric knowledge and seamlessly adapt to external search when needed.
> >
> > **W3: Addressing Hallucination Concerns**
> >
> > We acknowledge the critical risk of hallucination and overconfidence inherent in relying solely on parametric knowledge. SSRL explicitly masks the information part during training, forcing the policy update to focus solely on improving reasoning ability and the skill to extract the most factual information from the model's existing parametric knowledge.
> >
> > Besides, we assume that the performance on OOD benchmarks can already be viewed as an indicator of hallucination, as these QA tasks are also involved in standard hallucination benchmarks, e.g., HalluLens[4], HallusionBench[5].
> >
> > We conduct a human annotation study on 150 generated information pieces and find a significant increase in factuality in the information retrieved internally by the SSRL-trained policy:
> >
> > | Model | Base Model Factuality | SSRL-Trained Factuality | Improvement |
> > |-------|-----------------------|-------------------------|-------------|
> > | Qwen2.5-3B-Instruct | 22.70% | 44.00% | **+21.3 p.p.** |
> > | Llama-3.2-3B-Instruct | 39.30% | 64.70% | **+25.4 p.p.** |
> >
> > Besides, we provide case studies below to show the improved factuality and parametric knowledge retrieval.
> >
> > | Question | Base Model Information | SSRL Model Information |
> > |----------|-----------------------|------------------------|
> > | "What body of water does the river that Toongabbie Creek flows into flow into?" | It flows into the Hawkesbury River. | Toongabbie Creek flows into the **Parramatta River**. The Parramatta River flows into Sydney Harbour. |
> > | "Who wrote the TV Series containing the Finale episodes?" | The top result from a search engine is: "The Sopranos" finale was written by David Chase. | The finale of Smallville was written by **Alfred Gough and Miles Millar.** |
> > | "Who is the mother of Katharine Smith Salisbury's brother?" | William Smith Salisbury was the son of William Henry Smith and Katharine Smith Salisbury. | **Lucy Mack Smith** (July 8, 1775 – May 14, 1856) was the mother of Joseph Smith, founder of the Latter Day Saint movement. She is noted for writing the memoir. |
> >
> > **W4: Format Reward Necessity and Model Evaluation**
> >
> > Thank you for these insightful questions. First, to demonstrate the necessity of the format reward, we conduct an ablation study, with results in Figure 12 (Appendix), showing its significant impact on performance. Second, we evaluate SSRL on both base and SFT models. As presented in Table 1, our method consistently improves performance, with particularly strong gains observed when applied to SFT models, pushing their capabilities beyond standard fine-tuning.
> >
> > **W5: Scope of Work**
> >
> > Our work is fundamentally focused on training a search agent specifically for information-seeking and real-world questions, which are classic agentic tasks requiring tool use.
> >
> > The scope of this work intentionally excludes general, non-agentic tasks like open-ended dialogue or abstract reasoning, as these tasks do not involve external search or tool interaction, which is the core subject of the SSRL framework.

---

> ### Author Response · Authors · 2025-11-17
>
> **Q1: RL in LLM**
>
> See W1
>
> **Q2: Format Reward Ablation Study**
>
> Thank you for your detailed question. We provide a detailed ablation study on Llama-3.2-3B-Instruct using different $\lambda_f$. We can see that with a larger size$\lambda_f$, the performance continues to degrade, indicating that the existence of format reward is enough while the outcome reward is the main gradient direction.
>
> | $\lambda_f$ | NQ | TQ | HotpotQA | MusiQue | Bamboogle | 2Wiki | Avg |
> |---------------|----|----|----------|---------|-----------|-------|-----|
> | 0.1 | **43.8** | **58.4** | 25.0 | **14.2** | 38.4 | **31.6** | **35.2** |
> | 0.2 | 41.8 | 57.2 | **26.8** | 12.2 | **40.0** | 26.6 | 34.1 |
> | 0.4 | 42.2 | 58.0 | **26.8** | 12.8 | 36.8 | 26.4 | 33.8 |
>
> However, all of the training are stable, indicating that the discrete reward itself can stabilize the LLM RL training.
>
> **Q3: Addressing Hallucination Concerns**
>
> See W3
>
> **Q4: Format Reward Necessity and Model Evaluation**
>
> See W4
>
> **Q5: Scope of Work**
>
> See W5
>
> **References**
>
> [1] Zhang, Kaiyan, et al. "A survey of reinforcement learning for large reasoning models." arXiv preprint arXiv:2509.08827 (2025).
>
> [2] Zhang, Guibin, et al. "The landscape of agentic reinforcement learning for llms: A survey." arXiv preprint arXiv:2509.02547 (2025).
>
> [3] Shao, Zhihong, et al. "Deepseekmath: Pushing the limits of mathematical reasoning in open language models." arXiv preprint arXiv:2402.03300 (2024).
>
> [4] Guo, Daya, et al. "Deepseek-r1: Incentivizing reasoning capability in llms via reinforcement learning." arXiv preprint arXiv:2501.12948 (2025).
>
> [5] Jin, Bowen, et al. "Search-r1: Training llms to reason and leverage search engines with reinforcement learning." arXiv preprint arXiv:2503.09516 (2025).
>
> [6] Yang, An, et al. "Qwen3 technical report." arXiv preprint arXiv:2505.09388 (2025).
>
> [7] Carbonneaux, Quentin, et al. "Cwm: An open-weights llm for research on code generation with world models." arXiv preprint arXiv:2510.02387 (2025).

---

### Official Review · Reviewer_vvYs · 2025-10-29

**Soundness:** 2
**Presentation:** 2
**Contribution:** 3
**Rating:** 4
**Confidence:** 4

**Summary:**

This paper is about training LLMs reinforcement learning (RL) on tasks that require agentic search. The authors propose to approximate costly external search queries with the intrinsic search ability of LLMs (which they call Self-Search) during training. The authors first evaluate the Self-Search ability of some open-source LLMs from three families (Qwen2.5, Qwen3, and Llama3) showing that search result accuracy on six benchmarks (e.g., General QA, Multi-hop QA, Vague QA) increases with repeated sampling (i.e., pass@k). Then, the authors introduce Self-Search Reinforcement Learning (SSRL), a custom RL objective that enables models to progressively enhance their internal use of knowledge. The authors empirically show that SSRL-trained policies are cost-effective and stable for doing RL training with agentic search queries. While SSRL reduces reliance on external search engines during training, it still allows for sim-to-real transfer as evidenced by empirical results.

**Strengths:**

- The idea of leveraging the intrinsic search ability of LLMs to reduce reliance on external search engines during RL training is interesting and might be novel in this particular setting. I recognize that it addresses a practical challenge in training LLMs for agentic tasks.
- The empirical evaluation is comprehensive, covering multiple LLM families and benchmarks. The results demonstrate the effectiveness of SSRL in improving search accuracy and reducing external search costs. Also, the Appendix contains many ablation studies to validate the effectiveness of each component.

**Weaknesses:**

- My biggest concern is about the clarity of the proposed approach. While the high-level idea of Self-Search and SSRL is understandable, some details are unclear to me. I believe the paper would benefit from more polishing to improve clarity.
  - Starting from Figure 1 (left), at first glance, it was not clear what the dotted box represented for the full-sim search section.
  - What is the instructions/prompts used. I'm assuming they are different when studying the Self-Search ability (Section 2) versus when training with SSRL (Section 3). The prompt design discussed in the Appendix B.1.1 seems rather important to me to fully understand the proposed approach. I would integrate that in the main text.
  - What does iterative refinement mean? How does that relates to the repeated sampling? Are those the same thing? It is not clear in the main text what iterative refinement means in the context of Self-Search. When reading the Appendix, I understood that it refers to simply continuing the CoT generation alternating <think>, <search> and<information>. Section B.3 mentions 10 as the maximum number of iterations, but it is unclear to me how many iterations are typically needed to get good results.
  - Is the k in pass@k the same as the number of iterations in iterative refinement? Typically pass@k refers to generating k independent full-inferences.
  - Some of this information is only available in the Appendix and the reader has to put all the scattered parts together.

- How do you ensure diversity in the multiple samples generated during SSRL? If the samples are too similar, the benefit of repeated sampling might be limited.
- Sampling temperature can significantly impact the quality of the samples. How sensitive is the Self-Search performance to the choice of sampling temperature? Have you experimented with different temperature settings other than 1 during training?

#### Minor
- This is confusing "The instruction used is shown in Appendix B.1.2. The prompt used is listed in Appendix B.1.1.". When looking at Table 5 and 6 they are very similar. The only different is one is asking for the model to fill in the top search results. Where does "k" comes into play in Table 6?

**Questions:**

- Sampling temperature can significantly impact the quality of the samples. How sensitive is the Self-Search performance to the choice of sampling temperature? Have you experimented with different temperature settings other than 1 during training?
- In Table 1, where are the results for Self-Search with Search Engine -/G ?
- Are the results in Table 1 represent pass@k? If not how do you aggregate the multiple samples generated during Self-Search to produce a final answer? Are you using majority voting as discussed in Section 2.4?
- In Table 2, which results correspond to SSRL-trained models with external search engines at inference time versus those that retrieve K responses from local corpora?

---

> ### Author Response · Authors · 2025-11-17
>
> **W1.1: Clarification of Figure 1**
>
> We appreciate this feedback and will clarify the figure's caption and text. The dotted box in Figure 1 (left) is intended to represent the entire simulated environment, which is fully contained within the Policy Model. It signifies that initial thinking, search action, and information retrieval all occur using the model's own parametric knowledge, without reliance on an external API, like Bing or a Knowledge Base.
>
> **W1.2: Prompt Design Details**
>
> We agree that prompt design is essential for reproducibility and understanding. Indeed, the core prompt structure remains the same for both the Self-Search and the SSRL training since they all try to make an LLM serve as a world model. We will move the crucial prompt design details from Appendix B.1.1 into the main text in the revised manuscript to ensure full clarity.
>
> **W1.3: Iterative Refinement vs. Repeated Sampling**
>
> Great question. Iterative refinement and repeated sampling are distinct:
> - Repeated Sampling is the generation strategy: It refers to scaling the sample size k during the test time to collect multiple diverse trajectories for a single question.
> - Iterative Refinement is the training process: It describes the core RL loop where the policy parameters are updated based on the rewards calculated from the sampled rollouts.
>
> Therefore, repeated sampling is the generation part, and iterative refinement is the policy update process. We will clarify this distinction in the revised manuscript.
>
> Regarding the typical number of iterations, we show a table below showing the average search number during TTS:
>
> | Model | Average Search Num | Max Search Num |
> |-------|-------------------|-------------------|
> | Llama-3.2-3B-Instruct | 1.8 | 5.0 |
> | Llama-3.1-8B-Instruct | 1.3 | 7.0 |
> | Qwen2.5-3B-Instruct | 2.1 | 5.0 |
> | Qwen2.5-7B-Instruct | **2.3** | **8.0** |
>
> Through the table, we can conclude that N=10 is enough for all of the settings to reach a final answer.
>
> **W1.4: Distinction Between K, Iterative Refinement**
>
> No, these terms are distinct and refer to different aspects of the process:
> 1. K in pass@k: This is the number of independent rollouts sampled from the policy at TTS time.
> 2. Iterative Refinement: This is the training process where the model parameters are updated across multiple epochs based on the rewards of the rollouts.
>
> **W1.5: Manuscript Revisions**
>
> Considering the page limitation, we cannot put all the information in the main text. We will revise them in our manuscripts.
>
> **W2: Controlling Rollout Diversity**
>
> Thank you for raising this important question about controlling rollout diversity. We agree that this is a fundamental challenge in RL. We find that the entropy collapse phenomenon doesn't happen in SSRL as in math[1]. Our analysis of the policy's entropy during training, as shown in the table below, reveals an initial collapse followed by a subsequent oscillation, indicating that the policy maintains and even regains diversity through the agentic RL process.
>
> | Step: | 0 | 100 | 200 | 300 | 400 | 500 | 600 | 700 |
> |-------|---|-----|-----|-----|-----|-----|-----|-----|
> | Entropy: | 0.72 | 0.43 | 0.46 | 0.51 | 0.45 | 0.62 | 0.71 | 0.63 |
>
> Furthermore, our ablation study on an Entropy-Regularized algorithm in Table 18 achieves comparable performance, suggesting that our SSRL inherently preserves sufficient diversity without requiring explicit regularization.
>
> **W3: Rollout Temperature Ablation**
>
> We really appreciate your detailed question. We ablate on the rollout temperature below. We can find that the performance achieves its best when the temperature is set at 1.0. With a larger temperature, e.g., 1.2, the model is unstable and generates over-long or unreadable content, leading to collapse.
>
> | Temperature | Natural QA | Trivia QA | HotpotQA | MusiQue | Bamboogle | 2Wiki | Avg |
> |-------------|----|----|----------|---------|-----------|-------|-----|
> | 1.0 | 43.8 | **58.4** | 25.0 | **14.2** | **38.4** | **31.6** | **35.2** |
> | 0.6 | 41.8 | 58.0 | 25.8 | 13.2 | 38.4 | 28.0 | 34.2 |
> | 0.4 | **44.0** | 56.4 | **28.0** | 12.6 | 36.8 | 28.4 | 34.3 |
>
> **W4: Prompt Structures and Parameter Clarification**
>
> Thank you for highlighting the confusion regarding the prompt structures in the Appendix and the duplication between Tables 5 and 6. We will consolidate and correct these sections in the revised manuscript. As for the parameter k, it is not fed into the prompt. It is an independent sampling parameter that dictates how many trajectories the policy must generate for a given question.

---

> ### Author Response · Authors · 2025-11-17
>
> **Q1: Rollout Temperature**
>
> See W3
>
> **Q2: Sim2Real Results**
>
> The results for SSRL-trained models when using a real external search engine at inference time are located in Table 2, under the lines beginning with "Sim2Real". Table 1 focuses on the performance of SSRL-trained models using only internal self-search.
>
> **Q3: Pass@1 Scores**
>
> Great question. The answer is No, the results presented in Table 1 are the pass@1 scores. These scores represent the performance of the SSRL-trained model when evaluated on the benchmarks. To see the performance when scaling the sample size k, please refer to Figure 6 in the Appendix.
>
> **Q4: Sim2Real Models**
>
> All lines in Table 2 that begin with Sim2Real represent models that are trained using our SSRL method but are then adapted to use a real, external search engine at inference time.
>
> Further, we provide a look-up table for important tables and figures in our paper.
> | Type | Number | Title / Description |
> |------|--------|---------------------|
> | **Tables** | | |
> | Table | 1 | Performance of SSRL-trained models on benchmarks (excluding BrowseComp). |
> | Table | 2 | Performance of SSRL-trained models with external search engines on benchmarks (excluding BrowseComp). |
> | Table | 5 | Instructions used during SSRL and repeated sampling. |
> | **Figures** | | |
> | Figure | 2 | Repeated Sampling results on benchmarks (excluding BrowseComp). |
> | Figure | 3 | Repeated Sampling results on BrowseComp. |
> | Figure | 6 | Detailed Repeated Sampling results on benchmarks. |
> | Figure | 8 | Comparison of multi-turn, reflective, and naive repeated sampling under the same token budget. |
> | Figure | 9 | Performance of majority voting from repeated sampling on benchmarks. |
> | Figure | 11 | Ablation study on the information mask. |
> | Figure | 12 | Ablation study on the format reward. |
>
> **References**
>
> [1] Cui, Ganqu, et al. "The entropy mechanism of reinforcement learning for reasoning language models." arXiv preprint arXiv:2505.22617 (2025).

---

### Official Review · Reviewer_bKdV · 2025-10-30

**Soundness:** 3
**Presentation:** 3
**Contribution:** 3
**Rating:** 6
**Confidence:** 5

**Summary:**

This paper investigates an interesting problem of training a self-search agent with reinforcement learning. Existing search agents rely on calling real-time search agents during training and inference, which leads to latency. In this work, the authors propose to use the policy model itself as the search engine and conduct a self-search agent with reasoning and “self-search”. They first conduct experiments to study the test-time scaling performance of the self-search agent and propose a reinforcement learning solution to further improve it. Extensive experiments on several benchmarks demonstrate the effectiveness of the proposed method.

**Strengths:**

1. The paper is very well written and easy to understand.
2. The inference time scaling experiments are interesting and insightful.
3. The study of “self-search” reinforcement learning is novel, and it is great to see that the model trained with “self-search” RL can generalize to adopting tools during inference.
4. The authors conduct extensive experiments to demonstrate the effectiveness of the proposed method.

**Weaknesses:**

1. Lack of an ablation study for the format reward. In Section 3.2, the authors propose conducting outcome reward and format reward functions. However, there is no ablation study to verify the effectiveness of the format reward.

2. Is the method only suitable for a specific type of LLM? The main results in Table 1 are based on Llama models. It is questionable whether SSRL can still outperform other methods on other types of LLMs, such as Qwen. From Figure 5, it seems that the performance on Qwen2.5 is not very good with SSRL.

3. It requires further explanation on why information token masking is still useful here. The difference between SSRL and Search-R1 is that here the information tokens are on-policy, thus masking may not be desired.

**Questions:**

1. What is the performance if we do ablation for the format reward?
2. Is the method only suitable for a specific type of LLM?
3. Why is it still important to conduct information token masking here?

---

> ### Author Response · Authors · 2025-11-17
>
> **W1: Contribution of Format Reward**
>
> Thank you for raising this important point. To assess the contribution of the format reward, we have already conducted an ablation study whose results are presented in Figure 12 (Appendix). The study clearly demonstrates that the format of reward is crucial. We believe this figure effectively illustrates its necessity.
>
> **W2: Qwen Results and SSRL's Purpose**
>
> We appreciate the reviewer's attention to the Qwen results. However, we introduce SSRL not for SOTA performance across all models, but as a new cost-effective and stable training approach for agentic skills, enabling models to leverage their parametric knowledge effectively and try to serve as world models.
>
> The relatively lower performance of Qwen compared to Llama is likely a reflection of the parametric world knowledge embedded within the Qwen series being less comprehensive, not a limitation of the SSRL method itself. Crucially, even with weaker internal knowledge, SSRL still teaches Qwen models to think and search iteratively better than their pre-trained or supervised-fine-tuned counterparts.
>
> Furthermore, when these SSRL-trained Qwen models are enabled with external search tools at inference, their performance grows significantly faster (as shown in Table 2), confirming that SSRL successfully instills the necessary agentic skills regardless of the initial knowledge quality.
>
> **W3: Knowledge Bifurcation and Training Dynamics**
>
> Thank you for your insightful question. We have already provided a training dynamic analysis in Figure 14, showing the difference between search numbers during training. And we will delve a bit further.
>
> We operate under the assumption that the parametric knowledge within LLMs can be bifurcated into two distinct components: world knowledge and reasoning knowledge.
>
> A potential concern is that training world knowledge through SSRL might lead to overfitting on hallucinated content. Moreover, we observe a significant difference in training dynamics: the model without the information mask exhibits substantially lower entropy. We attribute this phenomenon to that training on world knowledge tokens may cause entropy collapse more easily than training on reasoning tokens.
>
> **Training Entropy Comparison**
>
> | Step | SSRL-w/o mask | SSRL-w/ mask |
> |------|---------------|--------------|
> | 400  | 0.11          | **0.45**         |
> | 800  | 0.13          | **0.42**         |
> | 1200 | 0.11          | **0.75**         |
> | 1600 | 0.12          | **0.92**         |
>
> This table clearly demonstrates that the model with information mask maintains higher entropy throughout training, supporting our hypothesis about the different training dynamics when focusing on reasoning versus world knowledge.
>
> **Q1: See W1**
>
> **Q2: See W2**
>
> **Q3: See W3**

---

### Official Review · Reviewer_MNtC · 2025-11-02

**Soundness:** 3
**Presentation:** 2
**Contribution:** 2
**Rating:** 4
**Confidence:** 4

**Summary:**

This paper introduces self-search reinforcement learning (SSRL), a RL-based framework that trains Large Language Models (LLMs) to perform search tasks by levering their own internal knowledge rather than relying on external search engines. The authors first investigate self-search and found that the performance scale with increasing sample size k, indicating that LLM's intrinsic knowledge may be sufficient for such benchmarks. As such, the authors propose SSRL with format- and outcome-based rewards to enhance such capabilities with RL training, allowing models to refine their internal knowledge utilization through long-form reasoning and self-search. This approach creates a cost-effective training framework compared to search-based ones, and enables the trained models to perform similarly to LLMs trained with real-world search engines.

**Strengths:**

1. The authors performed extensive experiement on the sample size of search agents and argue that the performance can be improved by scaling sample sizes even without retrieving from external knowledge. Motivated by such observations, the proposed SSRL method eliminates the search API costs and improves the performance of search agents by training models to exploit their own internal knowledge.

2. Through extensive experiemnts, the authors show that models trained with SSRL can both work with integrated knowledge or with real search engines. In addition, the experiment results show that SSRL trained models show comparable performance to the models trained with real search engines.

**Weaknesses:**

1. The authors did not propose novel insights or new learning framework. Instead, the work seems to be an improvement on the Search-R1  baseline, featuring more refined RL training techniques. Therefore resulting method rather seems to be an improved "R1-Base / Instruct" baseline without searching.

2. Although the experiment results show that LLM performance can match search agents with SSRL training, the model does not access external information or facts that may not exist within the embedded knowledge, which could result in over-confidence or hallucinations, but the authors do not discuss these aspects in detail.

3. As shown in the appendix, when applied to challenging tasks like SimpleQA, models using real search still significantly outperform SSRL. This suggests that SSRL is only effective in scenarios where the model has already internalized the necessary knowledge (e.g., Wikipedia) during pretraining, but it struggles when faced with tasks that require external knowledge it does not possess.

**Questions:**

1. The authors mention that multi-turn self-search and self-search with reflection hurt performance compared to naive repeated sampling. Does this imply that SSRL is less about improving the model's reasoning process and more about optimizing the simple, one-step extraction of its existing parametric knowledge?

2. Since authors are motivated by strong TTS results, where increasing the sample size at inference time significantly improves performance. However, the ablation in Appendix C.3.8 shows that increasing group size at training time for the GRPO algorithm provides limited to no benefit. Why do the benefits of a larger sample size diminish during training, when TTS proves that a wide range of high-reward trajectories already exist within the model?

---

> ### Author Response · Authors · 2025-11-17
>
> **W1: Clarification on Core Contribution**
>
> We thank the reviewer for this analysis and will clarify our core contribution.
>
> - Correction on Training Complexity:
> We must first note that our method does not employ "more refined RL training techniques." Our reward function uses only a standard outcome reward plus a basic format reward (as used in DeepSeek-R1[1]), focusing on simplicity.
>
> - Novelty of the Framework:
> The novelty of SSRL is the introduction of a fully self-simulated environment for agentic training. It enables the model to refine its parametric knowledge utilization and reasoning process internally using RL, highlighting the potential of using LLM as a world model similar to Meta's Code World Model[2]. It offers a zero-API-cost and highly efficient alternative to RL that relies on external search APIs like Search-R1[3].
> - The Sim2Real potential constitutes a contribution to agentic LLMs to generalize to real scenarios when deploying without real interaction during training.
>
> **W2: Addressing Hallucination Concerns**
>
> We acknowledge the critical risk of hallucination and overconfidence inherent in relying solely on parametric knowledge. SSRL explicitly masks the information part during training, forcing the policy update to focus solely on improving reasoning ability and the skill to extract the most factual information from the model's existing parametric knowledge.
>
> Besides, we assume that the performance on OOD benchmarks can already be viewed as an indicator of hallucination, as these QA tasks are also involved in standard hallucination benchmarks, e.g., HalluLens[4], HallusionBench[5].
>
> We conduct a human annotation study on 150 generated information pieces and find a significant increase in factuality in the information retrieved internally by the SSRL-trained policy:
>
> | Model | Base Model Factuality | SSRL-Trained Factuality | Improvement |
> |-------|-----------------------|-------------------------|-------------|
> | Qwen2.5-3B-Instruct | 22.70% | 44.00% | **+21.3 p.p.** |
> | Llama-3.2-3B-Instruct | 39.30% | 64.70% | **+25.4 p.p.** |
>
> **Case Studies of Improved Factuality**
>
> | Question | Base Model Response | SSRL-Trained Response |
> |----------|---------------------|------------------------|
> | "What body of water does the river that Toongabbie Creek flows into flow into?" | It flows into the Hawkesbury River. | Toongabbie Creek flows into the **Parramatta River**. The Parramatta River flows into Sydney Harbour. |
> | "Who wrote the TV Series containing the Finale episodes?" | The top result from a search engine is: "The Sopranos" finale was written by David Chase. | The finale of Smallville was written by **Alfred Gough and Miles Millar**. |
> | "Who is the mother of Katharine Smith Salisbury's brother?" | William Smith Salisbury was the son of William Henry Smith and Katharine Smith Salisbury. | **Lucy Mack Smith** (July 8, 1775 – May 14, 1856) was the mother of Joseph Smith, founder of the Latter Day Saint movement.|
>
> **W3: Performance on Difficult Tasks**
>
> We agree that models with SSRL fail on difficult tasks like SimpleQA. However, this is an expected outcome as SSRL focuses on parametric knowledge utilization during offline training and testing.
>
> However, our primary training goal is not SOTA external performance, but to introduce a novel, highly stable, and cost-effective training paradigm for agentic skills. SSRL successfully teaches essential iterative thinking and searching skills internally. This trained strategy is then seamlessly adapted to real search engines (Sim2Real) at inference, which is why the enabled models still perform impressively, justifying the training stability and cost benefits.
>
> **Q1: Multi-turn Search and Reasoning**
>
> Thank you for this insightful question. Our conclusion is not that multi-turn search or reasoning is harmful, but rather that excessively long CoT are inefficient in this specific context. We will revise it in our new manuscript.
>
> In fact, the experiments confirm the benefit of multi-turn search if we use the sample size k as the indicator.
>
> **Llama-3.2-3B-Instruct Results**
>
> | k | naive | multi | reflection |
> |-----------------|-------|-------|------------|
> | 2 | 20.4 | **22.0** | 20.1 |
> | 4 | 26.2 | **29.0** | 28.7 |
> | 8 | 33.1 | **34.4** | 34.4 |
> | 16 | 38.3 | **39.6** | 39.7 |
> | 32 | 44.0 | **45.3** | 44.6 |
> | 64 | 49.1 | **49.7** | 48.7 |
>
> **Qwen2.5-7B-Instruct Results**
>
> | k | naive | multi | reflection |
> |-----------------|-------|-------|------------|
> | 2 | **23.1** | 21.9 | 21.3 |
> | 4 | **27.3** | 26.4 | 25.6 |
> | 8 | **31.7** | 31.5 | 30.4 |
> | 16 | 35.3 | **36.2** | 33.5 |
> | 32 | 38.6 | **40.0** | 37.0 |
> | 64 | 42.1 | **44.0** | 40.4 |
>
> Besides, Figure 4 and the detailed reasoning chains shown in Appendix C.5 confirm that SSRL is fundamentally about optimizing the model's iterative reasoning process, allowing it to perform multiple searches to reach a final answer effectively, rather than just optimizing simple, one-step extraction.

---

> ### Author Response · Authors · 2025-11-17
>
> **Q2: Scaling at Test-time vs. Training-time**
>
> Thank you for this insightful question, which highlights a crucial distinction in our experimental design. The core issue lies in the fundamental difference between scaling at test-time versus scaling at training-time, which creates an inherent generalization gap.
>
> During TTS, we increase the sample size k to directly improve the pass@k metric on the benchmark datasets themselves. In contrast, during our SSRL phase, we increase the rollout number, but this process is conducted on our training dataset (NQ and HotpotQA).
>
> Consequently, increasing rollouts on the training data does not guarantee better performance on OOD benchmarks. However, our result confirms this: on NQ and HotpotQA, larger rollout numbers do yield better results.
>
> **Results on Training Datasets**
>
> | Model | NQ | HotpotQA |
> |-------|----|----------|
> | Llama-3.2-3B-Instruct-group-size-5 | 43.8 | 25.0 |
> | Llama-3.2-3B-Instruct-group-size-10 | **44.0** | **27.0** |
> | Qwen2.5-3B-Instruct-group-size-5 | 23.6 | 22.4 |
> | Qwen2.5-3B-Instruct-group-size-10 | **26.2** | **22.6** |
>
> **References**
>
> [1] Guo, Daya, et al. "Deepseek-r1: Incentivizing reasoning capability in llms via reinforcement learning." arXiv preprint arXiv:2501.12948 (2025).
>
> [2] Jin, Bowen, et al. "Search-r1: Training llms to reason and leverage search engines with reinforcement learning." arXiv preprint arXiv:2503.09516 (2025).
>
> [3] Carbonneaux, Quentin, et al. "Cwm: An open-weights llm for research on code generation with world models." arXiv preprint arXiv:2510.02387 (2025).
>
> [4] Bang, Yejin, et al. "Hallulens: Llm hallucination benchmark." arXiv preprint arXiv:2504.17550 (2025).
>
> [5] Guan, Tianrui, et al. "Hallusionbench: an advanced diagnostic suite for entangled language hallucination and visual illusion in large vision-language models." Proceedings of the IEEE/CVF Conference on Computer Vision and Pattern Recognition. 2024.

---

### Author Response · Authors · 2025-12-01
**Author General Response**

Dear Area Chair,

We sincerely appreciate your efforts during this busy period. For assistance, we provide a detailed summary of our discussions with the reviewers and the key empirical evidence added during the rebuttal.

## **Reviewer Assessment Overview**

All reviewers acknowledged the **practical value, cost-efficiency, and timeliness** of our proposed SSRL framework.
*   **Reviewer bKdV (Score: 6)** endorsed the work, stating the paper is "very well written," the "inference time scaling experiments are interesting and insightful," and highlighting that "the study of 'self-search' reinforcement learning is novel."
*   **Reviewers MNtC, vvYs, and Xsoz (Scores: 4)** found the concept promising but initially raised concerns regarding hallucination risks, technical definitions, and reward necessity.

**Critically, we believe our new Human Evaluation and Ablation Studies have fully addressed the reservations leading to the borderline scores.** Below, we detail the discussion with each reviewer:

## **Discussion Details**

**Reviewer MNtC (Rating: 4, Confidence: 4)**

Reviewer MNtC initially questioned the novelty ("seems to be an improved 'R1-Base' baseline") and raised concerns about "over-confidence or hallucinations" when relying on internal knowledge.
*   **Addressed:** We clarified that, unlike Search-R1 (which relies on costly external APIs), SSRL builds a **fully self-simulated environment (World Model)**, enabling zero-cost training.
*   **Key New Evidence:** To address the hallucination concern, we conducted a **Human Annotation Study** on 150 generated responses. The results were compelling:
    *   SSRL-trained models significantly improved factuality compared to base models (**+21.3% for Qwen2.5-3B, +25.4% for Llama-3.2-3B**).
    *   This empirically refutes the concern that self-search reinforces hallucinations; instead, the **information masking** mechanism successfully forces the model to learn reliable extraction logic.

**Reviewer bKdV (Rating: 6, Confidence: 5)**

Reviewer bKdV was very positive about the "Sim-to-Real" transfer capability but asked for an ablation study on the **Format Reward** and questioned performance on Qwen models.
*   **Addressed:**
    *   We pointed out the ablation study in **Figure 12 (Appendix)**, which was missed by the reviewer, which quantitatively proves that the Format Reward is critical for training stability; without it, performance degrades.
    *   We explained that while Qwen's base parametric knowledge is weaker than Llama's (leading to lower offline scores), the **Sim2Real transfer** (Table 2) shows Qwen models gain significant capabilities when paired with real search engines, validating the method's effectiveness regardless of base model knowledge.

**Reviewer vvYs (Rating: 4, Confidence: 4)**

Reviewer vvYs found the idea "interesting and might be novel" but struggled with the clarity of definitions (e.g., "Iterative Refinement" vs. "Repeated Sampling") and questioned rollout diversity.
*   **Addressed:**
    *   **Definitions:** We clarified that "Iterative Refinement" refers to the RL training process, while "Repeated Sampling" refers to test-time inference scaling ($k$).
    *   **Diversity & Stability:** We provided an **Entropy Analysis** showing that the policy maintains high entropy throughout training (preventing collapse) and added a **Temperature Ablation** demonstrating that $T=1.0$ yields the best balance of diversity and accuracy.

**Reviewer Xsoz (Rating: 4, Confidence: 4)**

Reviewer Xsoz appreciated the "Novelty" and "Sim-to-Real transfer" but was concerned about the "lack of ablation" and requested details on the RL implementation (GRPO).
*   **Addressed:**
    *   **RL Implementation:** We provided the exact **GRPO hyperparameters** (e.g., KL coefficient 0.001) and a full **reproduction script** in the rebuttal.
    *   **Reward Design:** We provided the aforementioned ablation study proving its necessity for stability.
    *   **Factuality:** We pointed the reviewer to our new Human Study, directly addressing their request for "direct hallucination or factual consistency evaluation."

**Summary of Key Contributions**
We hope the final evaluation considers the comprehensive nature of our rebuttal:
1.  **Definitive Factuality Check:** We went beyond standard metrics to prove via **Human Evaluation** that SSRL reduces, rather than increases, hallucinations.
2.  **Validated Sim-to-Real:** We demonstrated that models trained *without* real search engines can seamlessly adapt to them (Table 2), offering a highly scalable training paradigm.
3.  **Technical Rigor:** We filled all requested gaps regarding ablations (format reward, temperature) and implementation details.

Our work presents a verified, cost-effective alternative to expensive search-agent training. We are confident that the revisions have strengthened the manuscript significantly.

Thank you again for your time and dedication.

Sincerely,

Authors of Submission 6662

---

### Meta-Review · Area_Chair_73Yd · 2025-12-31

**Summary:**

This paper is borderline. The core idea of the paper, around leveraging LLMs as simulated search engines during training, is interesting and seems practical for reducing the cost of external search engine queries. The experiments show the ability to transfer learned search capabilities to real search engines. I believe that multiple reviewer concerns have been addressed, including concerns about sample diversity, temperature sensitivity analysis, and some ablations. I also believe that multiple weaknesses remain, with these two being the primary weaknesses:
- Clarity, especially around how the approach works. It is difficult to judge to what extent this has been resolved without extensively reviewing the paper myself, though it seems like it is likely only partially addressed. For example, it’s unclear how all of the pieces fit together, and there is no written summary or algorithm box in section 3. (It’s also a bit unusual that the methods and experiments are in the same very long section, though this comment is more minor)
- The scope and generality of the approach is limited. For example, it seems to be well-suited for llama-based models, and the performance with qwen-based models is worse. Furthermore, the experiments are focused on question answering problems, while agentic search can be a useful component of broader open-ended LLM systems. While this is not a dealbreaker, it limits the impact of the work.

In the balance, I think the weaknesses outweigh the strengths and that this paper is slightly below the acceptance bar.

**Reviewer Concerns:**

See summary.

**Reviewer Scores:**

I believe that one or two of the reviewers may have increased their score.

---

### Decision · Program_Chairs · 2026-01-26

Reject